

# Averaging Bias Correction for the Future Space-borne Methane IPDA Lidar Mission MERLIN

Yoann Tellier[1], Clémence Pierangelo[2], Martin Wirth[3], Fabien Gibert[1], Fabien Marnas[4]

[1]Laboratoire de Météorologie Dynamique (LMD/IPSL), CNRS, Ecole polytechnique, Palaiseau Cedex, France
[2]Centre National d'Etudes Spatiales (CNES), Toulouse CEDEX 9, France
[3]Deutsches Zentrum für Luft- und Raumfahrt (DLR), Oberpfaffenhofen, Weßling, Germany
[4]Capgemini Technology Services (for CNES), Suresnes, France

*Correspondence to*: Yoann Tellier (yoann.tellier@lmd.polytechnique.fr)

**Abstract.** The CNES/DLR project MERLIN is a future IPDA lidar satellite mission that aims at measuring methane dry-air mixing ratio columns ($X_{CH_4}$) in order to improve surface flux estimates of this key greenhouse gas. To reach a 1 % relative random error on $X_{CH_4}$ measurements, MERLIN signal processing performs an averaging of data over 50 km along the satellite trajectory. This article discusses how to process this horizontal averaging in order to avoid the bias caused by the non-linearity of the measurement equation with measurements affected by random noise and horizontal geophysical variability. Three averaging schemes are presented: averaging of columns of $X_{CH_4}$, averaging of columns of Differential Absorption Optical Depth (DAOD) and averaging of signals. The three schemes are affected both by statistical and geophysical biases that are discussed and compared and correction algorithms are developed for the three schemes. These algorithms are tested and their biases are compared on modeled scenes from real satellite data. To achieve the accuracy requirements that are limited to 0.2 % relative systematic error (for a reference value of 1780 ppb), we recommend performing the averaging of signals corrected from the statistical bias due to the measurement noise and from the geophysical bias mainly due to variations of methane optical depth and surface reflectivity along the averaging track. The proposed method is compliant with the mission relative systematic error requirements dedicated to averaging algorithms of 0.07% (±1 ppb for $X_{CH_4} = 1780$ ppb) for all tested scenes and all tested ground reflectivity values.

## 1    Introduction

Methane (CH₄) is the second most important anthropogenic greenhouse gas after carbon dioxide (CO₂) (IPCC, 2013). Despite its key role in global warming there are still uncertainties in the cause of the observed large fluctuations in the growth rate of atmospheric methane. Measuring atmospheric CH₄ concentration on a global scale with both high precision and accuracy is necessary to improve the surface flux estimate and thus, develop the knowledge of the global methane cycle (Kirschke et al., 2013; Saunois et al., 2016).

The Methane Remote Sensing Lidar Mission (MERLIN - website: https://merlin.cnes.fr/) is a joint French and German space mission with a launch scheduled for 2022 (Ehret et al., 2017). This mission is dedicated to the measurement of the integrated



methane dry-air volume mixing ratio ($X_{CH_4}$). The German Space Agency (DLR) is responsible for the payload while the French Space Agency (CNES) is responsible for the platform (MYRIADE Evolution product line). The Payload Data Processing center is under CNES responsibility with significant contribution from DLR. MERLIN's active measurement is based on an Integrated Path Differential Absorption (IPDA) lidar. This technique relies on the Differential Absorption Lidar (DIAL)

measurements of a space-borne laser. The emitted pulses are backscattered on earth's surface or dense clouds and measured by the instrument. Then, the column content of a specific trace gas along the line of sight is retrieved from the measured signals. As presented on Figure 2 differential absorption uses the difference in transmission between the on-line pulse with a frequency accurately set in the through of several $CH_4$ absorption lines and the off-line pulse which wavelength has a negligible $CH_4$ absorption (Ehret et al., 2008).

The MERLIN measurements require a well-defined processing that ensures the final performance of the mission. The processing chain is divided into four levels. Level 0 (L0) consists of raw data (backscattered signals and auxiliary data), level 1 (L1) processes the vertically resolved products and the Differential Absorption Optical Depth (DAOD) values for both individual signal shot pairs and for an horizontal averaging window. Level 2 (L2) computes the $X_{CH_4}$ for both individual signal shot pairs and for a horizontal averaging window, additionally using operational analyzes from Numerical Weather Prediction

(NWP) centers. Finally, level 3 (L3) produces $X_{CH_4}$ maps using a Kalman filter approach (Chevallier et al, 2017).

To reach a usable precision, space-borne IPDA lidar missions often require an averaging of measurements along the orbit's ground track (Grant et al. 1988). This process of averaging data horizontally is a general concern for IPDA lidar missions. The data processing of the ASCENDS mission (NASA carbon dioxide IPDA lidar mission) considers averaging of multiple lidar measurements along track over 10 seconds (70 km with no gaps) to reduce the random error on the carbon dioxide mixing

ratio: $X_{CO_2}$ (ASCENDS mission white paper, 2015). Likewise, MERLIN's averaging process is included into L1 and L2 algorithms in order to reduce the relative random error (RRE) of DAOD and $X_{CH_4}$ (Figure 1). For the MERLIN mission, measurements are averaged over a nominal window length of 50 km corresponding to about 150 shot pairs to reach a RRE of approximately 20 ppb.

The non-linearity of the equation relating signals and DAOD in combination with both the statistical noise inherent to any

measurement and the varying geophysical quantities (altitude, pressure, reflectivity) of the sounded scene increase the relative systematic error (RSE or bias) and impairs measurement accuracy. Werle et al. (1993) describes RRE reduction when averaging signal using the concept of Allan variance. Up to an optimal integration time, measurement variance reduces because the measurement is dominated by white noise. For greater integration times, the estimation is biased due to drifts inherent to the measurement systems. The aim of the present article is not to correct biases caused by real system drift but to correct biases

that are caused by the non-linearity of the IPDA lidar measurement equation. The former bias being negligible compared to the latter.

MERLIN must reach an unprecedented precision and accuracy on $X_{CH_4}$ with a targeted relative random error (RRE) of 1% (20 ppb). The targeted RSE must remain under 0.2% (±3 ppb) in 68% of cases, a limited budget of 0.07% (±1 ppb for a $X_{CH_4}$ of





1780 ppb) is allocated to biases introduced by averaging algorithms with algorithms to correct these averaging biases. To reach the RRE target, the level 1 and 2 of MERLIN's signal processing requires a horizontal averaging of data over 50 km along track (Kiemle et al., 2011). Thus, the single shot on-line and off-line random error is reduced by a factor of $\sqrt{150} \approx 12$. For instance, for the typical reflectivity (0.1), the on-line and off-line signal to noise ratios are of the order of 7 and 16 respectively

(resp.) and the equivalent SNRs for the averaged signals are resp. 78 and 192. This process greatly decrease the RRE of the $X_{CH_4}$.

Section 2 gives an overview of the IPDA equations and MERLIN data processing. Section 3 defines and compares biases of several averaging schemes (described below) and suggests correction algorithms. Section 4 presents a comparative evaluation of these averaging schemes and associated bias correction procedures using modeled scenes based on real satellite data. And

finally, in section 5, the results of the simulation are described and a "best approach" algorithm (i.e. the least biased on tested scenes) is proposed for MERLIN processing chain.

## 2    Overview of IPDA equations and the MERLIN processing chain

MERLIN active measurement is based on a short pulse Integrated Path Differential Absorption (IPDA) lidar. The column content of methane between the satellite and the "hard" target (ground, vegetation, clouds...) is retrieved by measuring the

light that is reflected by the scattering surface which is illuminated by two laser pulses with a slight wavelength difference. Figure 1 schematically shows the principle of the nadir-viewing space-borne lidar MERLIN. The pulse-pair repetition rate is 20 Hz and the sampling distance is 350 m considering a ground spot velocity of about 7 km/s. The on-line and off-line ground spots are separated by about 2 m which is negligible compared to the ground diameter of the spots of about 100 m (90% encircled energy). Shot-pairs will be averaged over a 50 km window (about 150 shots pairs). The on-line wavelength $\lambda_{\text{on}}$

(1645.552 nm; 6076.998 cm$^{-1}$) is positioned in the through of one of the methane absorption line multiplets whereas the off-line wavelength $\lambda_{\text{off}}$ (1645.846 nm; 6075.903 cm$^{-1}$), which serves as reference, is positioned so that the methane absorption is negligible (Figure 2). Both wavelengths are close enough so that interactions with the ground and the atmosphere and instrumental response can be considered identical, notably for reflectivity, which is defined as the ratio of the power reflected toward the satellite receiver to that incident on the "hard" target. The difference is thus mostly sensitive to the difference in

methane absorption.

For the sake of conciseness, we introduce for any variable $X$ the notation $X^{\text{on,off}}$ that represents interchangeably the on-line or off-line variables $X^{\text{on}}$ or $X^{\text{off}}$. Measuring the on-line and off-line pulse energies denoted $P^{\text{on,off}}$ (resp. $P^{\text{on}}$ or $P^{\text{off}}$), it is possible to compute the DAOD of methane, and then, retrieve $X_{CH_4}$ for the sounded column. Denoting $Q^{\text{on,off}}$ the measurements after normalization by the laser pulse energies, denoted $E^{\text{on,off}}$, and range $r$ which is the distance from satellite

to reflective target:

$$Q^{\text{on,off}} = \frac{P^{\text{on,off}} \cdot r^2}{E^{\text{on,off}}}. \qquad (1)$$





The DAOD used in this study, in which contribution of other gas are neglected, is denoted $\delta$ and is computed as (2):

$$\delta = \frac{1}{2} \cdot \ln\left(\frac{\varrho^{\text{off}}}{\varrho^{\text{on}}}\right) = -\frac{1}{2} \cdot \ln(\tau^2) \, , \tag{2}$$

where $\tau^2$ is the relative two-way transmission. From $\delta$, we can derive $X_{CH_4}$ from Eq. (3) (Ehret et al., 2008 ; Kiemle et al., 2011):

$$X_{CH_4} = \frac{\delta}{IWF} = \frac{\int_{p_{\text{surf}}}^{0} \text{vmr}_{CH4}(p) \cdot WF(p,T) \cdot dp}{\int_{p_{\text{surf}}}^{0} WF(p,T) \cdot dp} \, , \tag{3}$$

where $p_{\text{surf}}$ denotes the target pressure where the laser beam hits the ground, $p$ and $T$ are the pressure and temperature profiles and $\text{vmr}_{CH4}(p)$ is the dry-air volume mixing ratio profile of methane. The weighting function $WF(p,T)$ describes the measurement sensitivity of $X_{CH_4}$ along the vertical and $IWF$ is the integrated weighting function of the column. These quantities are computed from meteorological and spectroscopic data and the $WF$ is given by the following equation:

$$WF(p,T) = \frac{\sigma_{\text{on}}(p,T) - \sigma_{\text{off}}(p,T)}{g(p) \cdot (M_{air} + M_{H_2O} \cdot \rho_{H_2O}(p,T))} \, . \tag{4}$$

$M_X$ denotes the molecular masses of the chemical species $X$, $\rho_{H_2O}$ is the dry-air volume mixing ratio of water vapor, $g(p)$ strands for the acceleration of gravity – treated as altitude and hence p dependent – and here, $\sigma_{\text{on,off}}$ are the cross sections for the on-line or off-line wavelengths (not to be confused with the standard deviation notation $\sigma$ used elsewhere in this article). As previously mentioned, in order to reach the targeted 1% relative random error on $X_{CH_4}$ measurements, the signal processing

of MERLIN requires a horizontal averaging of data. However, we will show in next section that the non-linearity of Eq. (2) in combination with the measurement noise and the variability of the observed scene (surface elevation, reflectivity, meteorology) along the averaging window induces biases on the average $X_{CH_4}$.

### 3 Averaging schemes and bias correction: a theoretical approach

#### 3.1 Definitions

In the following, we will use the triangular bracket notation to denote the arithmetic sample mean $\langle Y \rangle = \frac{1}{N} \sum_{i=1}^{N_S} Y_i$ of the quantity $Y$ and $\Delta Y_i = Y_i - \langle Y \rangle$ will represent the deviation of the $i^{th}$ quantity to this arithmetic mean. By extension, when we use a weighted sample mean of the quantity $Y$, weighted by a quantity $Z$, we will denote it $\langle Y \rangle_{w[Z]} = \sum_{i=1}^{N_S} w_i[Z] \cdot Y_i$, where $w_i[Z] = Z_i / \sum_{k=1}^{N_S} Z_k$ are the normalized weights used. The expected value of an random variable $X$ will be denoted $E[X]$, and the fact that $X$ follows a normal distribution of mean value $\mu$ and variance $\sigma^2$ will be denoted $X \sim \mathcal{N}(\mu, \sigma^2)$.

We are interested in the retrieval of the column integrated methane concentration on a 50 km horizontal section along the satellite track. This quantity will be hereafter denoted $\overline{X_{CH_4}}^T$ (where $T$ stands for target). The information that we can compute using the satellite measurements is the shot-by-shot $X_{CH_4,i}$ ($i$ is the shot index) which is related to the shot-by-shot volume mixing ratio of methane $\text{vmr}_{CH_4,i}(p)$ and the shot-by-shot weighting function $WF_i(p)$ by Eq. (3). For the purpose of building





the data processing chains, all the quantities must be described on a gridded model (vertical and horizontal discretization) of the atmosphere. This grid is composed of $(N_l \cdot N_s)$ cells where $N_l$ is the number of vertical layers of the model and $N_s$ is the number of shots that we want to average along the satellite path. To model the atmosphere, the pressure at the interface of each layer (at each $N_l + 1$ levels) uses a hybrid-sigma coordinate system and are denoted $P_{i,j}$. Note that the standard notation for indices will be kept consistent throughout this article. The first index (often denoted $i$) will represent the shot index and the second index (often denoted $j$) will represent the layer index (or level index). The term "level" stands for the pressure vertical level. The pressure thickness of every layers, denoted $\Delta P_{i,j}$, is then derived from the pressure at every levels.

The discrete form of Eq. (3) is:

$$X_{CH_4,i} = \frac{\sum_{j=1}^{N_l} \text{vmr}_{CH4,i,j} \cdot WF_{i,j} \cdot \Delta P_{i,j}}{\sum_{j=1}^{N_l} WF_{i,j} \cdot \Delta P_{i,j}} \ . \tag{5}$$

In order to define the average value $\overline{X_{CH_4}}^T$, we must define average values for the volume mixing ratio of methane and the weighting function. As the two quantities are intensive properties, it is necessary to multiply them by the pressure thickness to get the corresponding additive quantity. The average volume mixing ratio and the average weighting function of the $j^{th}$ layer are thus given by:

$$\overline{\text{vmr}_{CH_4}}_j = \sum_{i=1}^{N_s} \pi_{i,j} \cdot \text{vmr}_{CH_4,i,j} \ , \tag{6}$$

$$\overline{WF}_j = \sum_{i=1}^{N_s} \pi_{i,j} \cdot WF_{i,j} \ , \tag{7}$$

where the weights are defined as:

$$\pi_{i,j} = \frac{\Delta P_{i,j}}{\sum_{k=1}^{N_s} \Delta P_{k,j}} \ . \tag{8}$$

The pressure thickness, as an extensive property, is averaged arithmetically and the average value is denoted $\overline{\Delta P}_j$. Then, we can define the average column integrated methane concentration as:

$$\overline{X_{CH_4}}^T = \frac{\sum_{j=1}^{N_l} \overline{\text{vmr}}_{CH4,j} \cdot \overline{WF}_j \cdot \overline{\Delta P}_j}{\sum_{j=1}^{N_l} \overline{WF}_j \cdot \overline{\Delta P}_j} \ . \tag{9}$$

### 3.2 Averaging schemes and types of biases

There are several ways to average the $X_{CH_4}$ provided the shot by shot normalized signals $Q_i^{\text{on,off}}$. Table 1 presents four different averaging schemes: averaging of columns of $X_{CH_4}$ (AVX – first line of Table 1) , averaging of columns of DAOD and IWF (AVD – second line of Table 1), averaging of signals (AVS – third line of Table 1) and averaging of quotients (AVQ – fourth line of Table 1). Since these four averaging schemes do not average the same physical quantity, they are differently biased.

There are two main causes of bias on the retrieved $X_{CH_4}$: the statistical bias and geophysical biases. The statistical bias which affects every shot individually is not produced by the averaging process and must be taken into account for shot by shot measurement. It is induced by the random nature of the measurement of on-line and off-line signals into non-linear equations.



Figure 3 illustrates the statistical bias, when on-line and off-line signals follow normal distributions. It highlights that, in this case, the DAOD derived from these signals is no longer normally distributed but skewed. The second main sources of bias are called geophysical biases. These biases are induced by the process of averaging. The successive averaged shots do not sound the same portion of atmosphere (surface pressure and gas concentrations vary), they are not reflected on the same surface

(reflectivity varies) and the elevation of the scattering surface is not constant in general (altitude and hard target surface pressure vary). All these variations of geophysical quantities induce several biases on the average values.

The first scheme, AVX, directly averages the column mixing ratios of methane. Every shot is impacted by the statistical bias developed in section 3.3.1. Furthermore, since a column with a high total molecular content and another with less molecules would count the same in the averaged mixing ratio, the uniform weighting of methane concentrations leads to the creation of

a bias that is called geophysical bias of type 1 described in section 3.4.1.

The second scheme, AVD, computes the ratio of the mean DAOD and the mean IWF. It is also impacted by the statistical bias (cf. section 3.3.1). However, this scheme takes into account the fact that every column does not present the same molecular content as DAOD and IWF are averaged separately. Thus, it is not impacted by geophysical bias of type 1.

The third scheme, AVS, averages signals before computing relative transmissions, DAOD and $X_{CH_4}$. The statistical bias is

only applicable to the resulting average signals such that, this bias is highly reduced compared to AVX and AVD (cf. section 3.3.2). However, geophysical biases are increased. First, when the DAOD varies from shot to shot – due to altitude (or surface pressure) variations or methane concentration variations for instance – the DAOD computed from average signals is not representative of the true mean DAOD. This is called geophysical bias of type 2 presented in section 3.4.2. Secondly, for the AVS scheme, the average DAOD is weighted by the off-line signal strength. Consequently, this scheme is more sensitive to

the less noisy measurements which, on the one hand, implies that the variance of average quantities is lower but, on the other hand, a correlation between methane concentration and reflectivity implies a bias. This is called geophysical bias of type 3 also discussed in section 3.4.2.

The fourth scheme, AVQ, averages transmissions before computing average DAOD (and average $X_{CH_4}$). The transmissions for every column are averaged with a uniform weighting. Note that the major drawback of this scheme is that it mixes several

bias sources that cannot be easily corrected. Indeed the averaging being made inside the logarithm, it is not possible to separate into two terms the bias due to the measurement noise and the variation of geophysical parameters of the scene (cf. Appendix **Erreur ! Source du renvoi introuvable.**). This scheme gives very bad performances and will not be considered in the next sections.

In the following, for each averaging scheme of Table 1 (except averaging of quotients), we will quantify separately the

statistical bias and the geophysical biases and will in the end combine them in order to determine the total bias for various scenarios.





### 3.3    Statistical bias

#### 3.3.1    Statistical bias on AVX and AVD

The averaging of columns (either $X_{CH_4}$ or DAOD) needs DAODs to be computed for every couple of measurements

$(Q^{\text{off}}, Q^{\text{on}})$. However, as the measurements are affected by random noise and the IPDA lidar equation (Eq. (2)) is not linear, a

bias appears when computing the DAOD (Figure 3). Let's define the estimator of the DAOD $\hat{\delta}$ as follows:

$$\hat{\delta} = \frac{1}{2} \cdot \ln\left(\frac{Q^{\text{off}}}{Q^{\text{on}}}\right) . \tag{10}$$

The total noise contributions affecting off-line and on-line signals are statistically independent. Thus, for each single shot, $Q^{\text{on}}$

and $Q^{\text{off}}$ can be considered as independent random variables. Furthermore, due to the relatively high number of photons in a

single pulse, we can assume that these random variables are normally distributed around a mean value $\mu^{\text{on,off}}$ and with a

standard deviation $\sigma^{\text{on,off}}$. Then, Eq. (10) can be decomposed into three terms:

$$\delta = \frac{1}{2} \cdot \ln\left(\frac{\mu^{\text{off}} + \sigma^{\text{off}} . X^{\text{off}}}{\mu^{\text{on}} + \sigma^{\text{on}} . X^{\text{on}}}\right) = \frac{1}{2} \cdot \ln\left(\frac{\mu^{\text{off}}}{\mu^{\text{on}}}\right) + \frac{1}{2} \cdot \ln\left(1 + \frac{X^{\text{off}}}{SNR^{\text{off}}}\right) - \frac{1}{2} \cdot \ln\left(1 + \frac{X^{\text{on}}}{SNR^{\text{on}}}\right), \tag{11}$$

where $X^{\text{on,off}}$ follow standard normal distributions. And the signal-to-noise ratios are defined as:

$$SNR^{\text{on,off}} = \frac{\mu^{\text{on,off}}}{\sigma^{\text{on,off}}} . \tag{12}$$

The first term of Eq. (11) is the parameter that needs to be estimated (i.e. the unbiased DAOD) and the two last terms are error

terms that correspond to the bias of $\hat{\delta}$ due to the non-linearity of the function:

$$Bias_{\text{stat}}(\hat{\delta}) = \frac{1}{2} \cdot E\left[\ln\left(1 + \frac{X^{\text{off}}}{SNR^{\text{off}}}\right)\right] - \frac{1}{2} \cdot E\left[\ln\left(1 + \frac{X^{\text{on}}}{SNR^{\text{on}}}\right)\right] . \tag{13}$$

The task is now to evaluate this bias to remove, or at least reduce it. Analytically, under the normal distribution hypothesis,

the expected values are defined by:

$$E\left[\ln\left(1 + \frac{X^{\text{on,off}}}{SNR^{\text{on,off}}}\right)\right] = \frac{1}{\sqrt{2\pi}} \int_{-SNR^{\text{on,off}}}^{+\infty} \ln\left(1 + \frac{x}{SNR^{\text{on,off}}}\right) \cdot e^{-\frac{x^2}{2}} \mathrm{d}x . \tag{14}$$

Providing that $SNR^{\text{on,off}}$ are high enough, we can use a Taylor expansion of the logarithm around zero so that the bias can be

approximated by the following formula (Bösenberg, 1998):

$$Bias_{\text{stat}}(\delta) \approx \frac{1}{4}\left[\frac{1}{(SNR^{\text{on}})^2} - \frac{1}{(SNR^{\text{off}})^2}\right] . \tag{15}$$

The assumption that the signals follow a normal distribution does not rigorously hold when the DAOD is computed. Indeed,

over dark surfaces (low reflectivity), the SNR may happen to be so low that either one or both signals $Q^{\text{off}}$ and $Q^{\text{on}}$ takes

negative values, hence, DAOD is undefined. This can actually happen as $Q^{\text{on,off}}$ correspond to the photon count from which

the background has been subtracted. Whenever one of the signals is negative, the corresponding couple $(Q^{\text{off}}, Q^{\text{on}})$ must be

discarded. This is equivalent to considering equations (11) and (13) with $X^{\text{on,off}}$ as a left-truncated normal distribution with a

mean value of zero, a variance of one and a left-truncation at $-SNR^{\text{on,off}}$ (Johnson et al., 1994). When done so it comes that:





$$E\left[\ln\left(1+\frac{X^{\text{on,off}}}{SNR^{\text{on,off}}}\right)\right] = \frac{1}{\sqrt{2\pi}\left[1-\Phi(-SNR^{\text{on,off}})\right]}\int_{-SNR^{\text{on,off}}}^{+\infty}\ln\left(1+\frac{x}{SNR^{\text{on,off}}}\right)\cdot e^{-\frac{x^2}{2}}\mathrm{d}x \,, \qquad (16)$$

where $\Phi$ is the standard normal cumulative distribution function.

To correct the bias due to the non-linearity of the IPDA lidar equation, the SNR must be estimated. Once done, the bias correction scheme would either need to estimate the bias directly from the approximate Taylor expansion formula of Eq. (15)

or estimate the bias using Eq. (13) and a numerical computation of Eq. (16). Typically, for MERLIN observations, the error made by the Taylor expansion of Eq. (15) instead of using Eq. (16) is lower than 1 ppb on the $X_{CH_4}$ for a surface reflectivity value greater than 0.1 ($SNR^{\text{off}} \approx 16$ and $SNR^{\text{on}} \approx 7$) as shown on Figure 4. Table 2 shows the error made by using the Taylor expansion instead of computing the truncated normal integral. For values of reflectivity smaller than 0.1, it would be preferable to use the exact formula for the bias presented in equations (13) and (16). Further study (not presented here) shows that for

very low reflectivity, the estimation of the noise induced bias is really sensitive to an error on the SNR and this correction is no longer applicable in practice. The way statistical bias on the DAOD is translated to bias on $X_{CH_4}$ will be treated in section 3.5.

### 3.3.2    Statistical bias on AVS

The third averaging scheme defined on Table 1, AVS, averages on-line and off-line signals separately. The corresponding

estimator of the average DAOD is written:

$$\hat{\delta}^{avs} = \frac{1}{2}\cdot\ln\left(\frac{\langle Q^{\text{off}}\rangle}{\langle Q^{\text{on}}\rangle}\right). \qquad (17)$$

Consistently with section 3.3.1, we consider the individual signals to be normal random variables of mean $\mu_i^{\text{on,off}}$ and standard deviation $\sigma_i^{\text{on,off}}$. The parameters of the distributions depend on the shot $i$ since each shot is considered as the realization of a different distribution depending on the geophysical parameters of the scene (reflectivity, atmospheric transmission, surface

pressure). The successive measurements are considered independent and, as the sum of independent normal random variables, is a normal random variable. We introduce $S^{\text{on,off}}$ the average random variable:

$$S^{\text{on,off}} = \langle Q^{\text{on,off}}\rangle \sim \mathcal{N}\left(m^{\text{on,off}},\left(\epsilon^{\text{on,off}}\right)^2\right), \qquad (18)$$

where the mean and variance of $S^{\text{on,off}}$ are:

$$m^{\text{on,off}} = \langle\mu^{\text{on,off}}\rangle\,, \qquad (19)$$

$$\left(\epsilon^{\text{on,off}}\right)^2 = \frac{1}{N^2}\sum_{i=1}^{N}\left(\sigma_i^{\text{on,off}}\right)^2. \qquad (20)$$

The empirical estimate of the SNR of the equivalent measurement $S^{\text{on,off}}$ on the whole averaging window can be written:

$$SNR_{\text{eq}}^{\text{on,off}} = \frac{m^{\text{on,off}}}{\epsilon^{\text{on,off}}} = \left(\sum_{i=1}^{N}\mu_i^{\text{on,off}}\right)\left(\sum_{i=1}^{N}\left(\frac{\mu_i^{\text{on,off}}}{SNR_i^{\text{on,off}}}\right)^2\right)^{-\frac{1}{2}}. \qquad (21)$$

Given these definitions, we can write the bias due to shot random variations as in Eq. (13):





$$Bias_{stat}(\delta^{avs}) = E[\delta^{avs}] - \frac{1}{2} \cdot \ln\left(\frac{m^{off}}{m^{on}}\right) = \frac{1}{2} \cdot E\left[\ln\left(1 + \frac{\chi^{off}}{SNR_{eq}^{off}}\right)\right] - \frac{1}{2} \cdot E\left[\ln\left(1 + \frac{\chi^{on}}{SNR_{eq}^{on}}\right)\right], \tag{22}$$

Provided an estimation of the shot-by-shot SNR, we can estimate the bias of Eq. (22) with the same methods than in section 3.3.1, either considering the simplified Taylor expansion approximation (Eq. (15)), or the more accurate integral of truncated normal distribution (Eq. (13) and (16)). Compared to AVX and AVD schemes, the equivalent SNRs, when averaged

horizontally on 150 shot pairs, are considerably larger and as a consequence, the bias is considerably smaller. The Taylor expansion approximation holds really well and an error on the estimation of SNR has a negligible impact on the bias estimation.

### 3.4 Geophysical bias

#### 3.4.1 Geophysical bias of type 1 on AVX

Considering an arithmetic averaging for both AVX and AVD schemes yields different results, since the former scheme

averages concentrations and the latter averages quantities that are proportional to number of molecules of methane. Whereas the AVX scheme computes the arithmetic mean of $X_{CH_4}$ (Eq. (23)), the AVD scheme computes average $X_{CH_4}$ weighted by the IWF (Eq. (24)):

$$\overline{X_{CH_4}}^{avx} = \left\langle \frac{\delta}{IWF} \right\rangle = \langle X_{CH_4} \rangle, \tag{23}$$

and

$$\overline{X_{CH_4}}^{avd} = \frac{\langle \delta \rangle}{\langle IWF \rangle} = \langle X_{CH_4} \rangle_{w[IWF]}, \tag{24}$$

where

$$w_i[IWF] = \frac{IWF_i}{\sum_{k=1}^{N_s} IWF_k}. \tag{25}$$

For the AVX scheme, the quantity that is averaged is the column concentration which is an intensive property. If a uniform weighting is considered, there is the same contribution from columns with many molecules as from ones with less molecules.

For this scheme, a variation of IWF from shot to shot (i.e. variation of altitude and/or surface pressure) leads to an overestimation of the methane content of columns that contain fewer molecules in the average $X_{CH_4}$. This bias will be called geophysical bias of type 1 and is simply corrected by introducing the weighted average by the IWF. This has to be taken into account when computing the statistical bias for this scheme as will be introduced in section 3.5.

On the contrary, the AVD scheme averages the extensive properties of DAOD and IWF separately. Thus, when the DAODs

are averaged, the molecule amount is preserved such that the AVD scheme is not affected by a type 1 geophysical bias.

#### 3.4.2 Geophysical bias of type 2 and 3 on AVS

Once the bias induced by the random nature of the measurement has been subtracted the resulting estimator is still biased by the effects of horizontal variations of geophysical quantities. Indeed, using Eq. (11), we are left with:

$$\overline{\delta^{avs}} = \delta^{avs} - Bias_{stat}(\delta^{avs}) = \frac{1}{2} \cdot \ln\left(\frac{m^{off}}{m^{on}}\right), \tag{26}$$



where $m^{\text{off}}$ and $m^{\text{on}}$, as defined by Eq. (19), are the average of signal expected values. Successive shot-pairs are not sounding the same column of atmosphere, such that altitude, reflectivity and CH$_4$ concentration vary horizontally. Unlike for the AVX and AVC schemes where the ratio is computed separately, for the AVS scheme, the changing reflectivity or atmospheric transmission does not cancel out directly when computing the ratio of signals. Although measurement random noise is

significantly reduced, a geophysical noise appear. We can rewrite Eq. (26) as:

$$\overline{\delta^{avs}} = -\frac{1}{2} \cdot \ln\left(\sum_{i=1}^{N_s} \frac{\mu_i^{\text{off}}}{\sum_{k=1}^{N} \mu_k^{\text{off}}} \cdot \exp(-2 \cdot \delta_i)\right) = -\frac{1}{2} \cdot \ln\left(\sum_{i=1}^{N_s} w_i[\mu^{\text{off}}] \cdot \tau_i^2\right). \tag{27}$$

Using a Taylor expansion of Eq. (27), it is possible to show that $\overline{\delta^{avs}}$ approximately equals the mean of the single-shot DAODs weighted by the $w_i[\mu^{\text{off}}]$:

$$\overline{\delta^{avs}} = \sum_{i=1}^{N_s} w_i[\mu^{\text{off}}] \cdot \delta_i + R_{\text{res}}, \tag{28}$$

where $R_{\text{res}}$ is the residual error of the linear approximations when averaging DAODs instead of transmissions. This term will be called type 2 geophysical bias. Equations (27) and (28) lead to:

$$R_{\text{res}} = -\frac{1}{2} \cdot \ln\left(\sum_{i=1}^{N_s} w_i[\mu^{\text{off}}] \cdot \exp(-2 \cdot \delta_i)\right) - \sum_{i=1}^{N} w_i[\mu^{\text{off}}] \cdot \delta_i. \tag{29}$$

Note that when the DAOD is constant all along the averaging scene, $R_{\text{res}}$ is exactly zero. Furthermore, when $\mu^{\text{off}}$ is horizontally constant, $R_{\text{res}}$ is approximately the variance of the DAOD. In fact, the term $R_{\text{res}}$ is twofold: on the one hand, it is

linked to DAOD fluctuations and, on the other hand, to the correlation between DAOD and reflectivity fluctuations. These correlations might occur for instance if there are covariations between topography (and thus DAOD) and surface type (e.g. snow – low reflectivity - over mountain tops – low surface pressure and thus low DAOD). In the general case, $R_{\text{res}}$ is not zero and can be estimated using $\overline{\delta^{avs}}$ from Eq. (26), corrected for the statistical bias only, to compute a first order estimate for $X_{CH_4}$:

$$X_{CH_4}^{(1)} = \frac{\overline{\delta^{avs}}}{\langle IWF\rangle_{w[Q^{\text{off}}]}}, \tag{30}$$

Using $Q^{\text{off}}$ instead of $\mu^{\text{off}}$ which is unknown, and estimating $R_{\text{res}}$ as:

$$R_{\text{res}}^{(1)} \approx -\frac{1}{2} \cdot \ln\left(\sum_{i=1}^{N_s} w_i[Q^{\text{off}}] \cdot \exp\left(-2 \cdot X_{CH_4}^{(1)} \cdot IWF_i\right)\right) - \overline{\delta^{avs}}. \tag{31}$$

This process could be turned into an iterative correction. However, the first order estimate is sufficiently accurate in all cases (not shown).

According to Eq. (28), we notice that the AVS scheme, corrected for type 2 geophysical bias, computes an average DAOD

weighted by the off-signal strength. Since the main cause of variation of the off-line received power is the variation of surface or hard target reflectivity, the transmissions associated to brighter scenes count more in the average than the transmissions of darker scenes. The AVS scheme averages the measurements in such a way that a greater weight is given to high SNR signals. Consequently, this DAOD estimate is more precise (lower standard deviation) but also biased. This bias is called type 3 geophysical bias and will be defined in section 3.5.



### 3.5 From biases on DAOD to biases on $X_{CH_4}$

In sections 3.3 and 3.4, the statistical and geophysical biases on DAOD have been derived. Here we are interested in translating biases on DAOD to biases on $X_{CH_4}$ that we want to estimate. As shown by Eq. (3), $X_{CH_4}$ is obtained by dividing the DAOD by the IWF. This needs the IWF to be averaged horizontally consistently with the DAOD averaging scheme. Not only, the

computation of the average IWF with consistent weights is important to compute $X_{CH_4}$, but it is also needed by the data users for the assimilation to transport models.

For the AVD scheme, the DAODs are arithmetically averaged with a uniform weight. Hence, the IWF must be averaged in the same fashion. A shot-by-shot DAOD bias according to Eq. (13) translates into a statistical bias on $X_{CH_4}$ as follows:

$$Bias_{\text{stat}}\left(\overline{X_{CH_4}}^{avd}\right) = \frac{\langle Bias_{\text{stat}}(\widehat{\delta})\rangle}{\langle IWF\rangle} . \tag{32}$$

For the AVX scheme, $X_{CH_4}$ is computed for every shot. The statistical bias on every shot is the quotient of the bias on the shot DAOD over the shot IWF. However, when horizontally averaging the statistical bias on $X_{CH_4}$, the type 1 geophysical bias must be taken into account (section 3.4.1). The average bias should be weighted by the shot-by-shot IWF as in Eq. (24):

$$Bias_{\text{stat}}\left(\overline{X_{CH_4}}^{avx}\right) = \left\langle \frac{Bias_{\text{stat}}(\delta)}{IWF}\right\rangle_{w[IWF]} . \tag{33}$$

For the AVS scheme, the IWF must be weighted consistently with the averaging scheme. Equation (28) shows that the average

DAOD is weighted by the off-line signal strength. As presented in third line of Table 1, in order to keep the mixing ratio of methane consistent, the averaging of the IWF must also be weighted by $w_i[Q^{\text{off}}]$. Consistently, the translation of bias on the DAOD to bias on the $X_{CH_4}$ considers the same weighting for IWF. The statistical bias translates from Eq. (22) to:

$$Bias_{\text{stat}}\left(\overline{X_{CH_4}}^{avs}\right) = \frac{Bias_{\text{stat}}(\delta^{avs})}{\langle IWF\rangle_{w[Q^{\text{off}}]}} . \tag{34}$$

Concerning geophysical biases, a type 2 geophysical bias (due to the linearization of DAOD variations and the correlation of

signal and transmission fluctuations) described by Eq. (31) becomes:

$$Bias_{geo2}\left(\overline{X_{CH_4}}^{avs}\right) = \frac{R_{res}^{(1)}}{\langle IWF\rangle_{w[Q^{\text{off}}]}} . \tag{35}$$

The geophysical bias of type 3, caused by the higher sensitivity to the spots with higher reflectivity, could be written as:

$$Bias_{geo3}\left(\overline{X_{CH_4}}^{avs}\right) = \frac{\langle\delta^{\text{true}}\rangle_{w[Q^{\text{off}}]}}{\langle IWF\rangle_{w[Q^{\text{off}}]}} - \frac{\langle\delta^{\text{true}}\rangle}{\langle IWF\rangle} . \tag{36}$$

Indeed, the AVS scheme does not measure the "true" concentration of $CH_4$ on the 50 km window. The weighting by $w_i[Q^{\text{off}}]$

implies that greater weight is granted to shots measuring brighter targets. This could be detrimental to the measurement if there were a strong correlation between reflectivity and $CH_4$ concentration on a global scale, which should not be the case. For assimilation or inverse modelling to models with a higher resolution than 50 km, the weighting could also be taken into account in the forward model for the XCH4.



## 4 Methodology to test averaging algorithms and their bias corrections

### 4.1 Data sets (latitude, longitude, altitude, surface pressure, relative reflectivity)

The three averaging schemes and their associated bias will be tested on scenes modeled from real satellite data in terms of geophysical properties. For this purpose, we are interested in simulating the signals $Q_i^{on,off}$ and the integrated weighting function $IWF_i$, both on a 50 km scale. To be computed, the signals require the weighting functions for every shot ($WF_{i,j}$), the volume mixing ratio of methane ($vmr_{CH_4,i,j}$), both defined on the pressure grid ($P_{i,j}$), and the target reflectivity ($\rho_i$) for every shot. The integrated weighting function is computed from $WF_{i,j}$ (and $P_{i,j}$). The data sets are built from satellite data provided by the SPOT-5 satellite for latitude, longitude and relative reflectivity, the Shuttle Radar Topography Mission (SRTM) digital elevation map data for topography and European Centre for Medium-Range Weather Forecasts (ECMWF) analyses for surface pressure from which we deduce the pressure grid on 150 shots and 19 levels.

SPOT-5 was a CNES satellite launched in 2002 and operated until 2015 (Gleyzes et al, 2003). Amongst the 5 spectral bands of the High Resolution Geometric (HRG) instrument, it has a spectral band in the Short Wave InfraRed domain (1.55 to 1.7 µm) with a spatial resolution of 20 m. This band includes the MERLIN laser wavelength and, as we expect spectral variations of surface albedo to have rather low spectral variations, we use the "Spot SYSTEM SCENE level 1A" product (images using radiometric corrections, equivalent radiance in W.m⁻².Sr⁻¹.µm⁻¹) as a proxy of surface reflectivity. Indeed, as we were careful to select images with no clouds, we neglect the effect of atmospheric extinction on the SPOT-5 measurements. Note that we are interested here in a description of the reflectivity variations in the 50 km averaging window, not by the absolute value of reflectivity. This is why we consider this SPOT-5 product as suitable, and we will anyway scale it to any prescribed value of surface reflectivity in the simulations described hereafter. The topography is taken from the SRTM digital elevation model (Jarvis et al, 2008), which has a spatial resolution of about 90 m. Surface pressure is taken from ECMWF 4D variational analyses from the long window daily archive and interpolated at SRTM grid points. A correction from difference between ECMWF Integrated Forecasting System (IFS) model topography and SRTM altitude is applied. In order to make both SRTM and SPOT-5 data consistent, the three selected SPOT-5 images are first process by a low pass convolution, to obtain a 90 m spatial resolution, and then projected into the SRTM geometry. Note that the spatial resolution thus obtained is also close to MERLIN single shot footprint. Table 3 summarizes the data sets content and resolutions.

Three sites have been selected to be representative of topographic variability; they are located in the neighborhood of three French cities: Toulouse, Millau and Chamonix. The different characteristics of the three samples are described in Table 4. Figure 5 and Figure 6 show the variation of surface pressure and relative variations of reflectivity along the averaging scheme. Toulouse presents a medium variation of geophysical parameters (altitude and thus surface pressure), Millau presents a high variation and Chamonix a very high variation. Figure 7 shows the global cumulative distribution of standard deviations of altitude of SRTM database worldwide. We notice that 67%, resp. 97% of the scenes present a lower altitude standard deviation than the one considered on the Millau, resp. Chamonix data.





For sensitivity study purpose, the reflectivity relative variations from the SPOT-5 data set are multiplied by a reference mean reflectivity that can be chosen to obtain the usable scene reflectivity. Four mean reflectivity values will be considered: 0.1 (vegetation), 0.05 (mixed water/vegetation), 0.025 (sea/ocean), 0.016 (ice/snow).

The pressure grid $P_{i,j}$ and the pressure thickness grid $\Delta P_{i,j}$ are obtained from surface pressure $P_i^{surf}$ from ECMWF analyses data set using a hybrid-sigma coordinate system.

The methane volume mixing ratio, $\mathrm{vmr}_{CH_4,i,j}$, is arbitrarily set to values assumed to be realistic. For every shot $i$ and layer $j$:

$$\begin{cases} \mathrm{vmr}_{CH_4,i,j} = 1780 \text{ ppb} & \text{if } P_{i,j} < (\max(p_i^{surf}) - \min(p_i^{surf}))/2 \\ \mathrm{vmr}_{CH_4,i,j} = 1880 \text{ ppb} & \text{otherwise} \end{cases} \qquad (37)$$

This replicates the possible correlation between methane concentration and altitude (more methane in valleys and less over mountain tops).

Finally, the weighting functions are calculated, as described in Eq. (4), from methane absorption cross-sections and meteorological data ($\Delta P_{i,j}$, temperature, humidity). They are computed using $CH_4$ absorption cross-sections from the 4A radiative transfer model (Scott and Chédin, 1981; Chéruy et al., 1995) on a reference winter mid-latitude atmosphere from the Thermodynamic Initial Guess Retrieval (TIGR) data set (Chevallier et al., 2000). The sensitivity to the thermodynamic condition of the atmosphere has been tested and is negligible here (not shown).

## 4.2 Overall test framework

The aim of the simulation is to compare the biases of the estimated $X_{CH_4}$ for several averaging schemes and to evaluate the accuracy of the bias correction. A global description of the simulation is presented on Figure 8. Each simulation case considers a typical number of $N_s = 150$ double-shots per averaging window, approximately corresponding to 50 km along the satellite ground track. It relies on a description of the geophysical scene in terms of surface pressure $P_i^{surf}$, reflectivity $\rho_i$ , an arbitrary $CH_4$ concentration field $\mathrm{vmr}_{CH_4,i,j}$, and weighting functions $WF_{i,j}$ (cf. section 4.1). Then, the on-line and off-line signals are computed from surface reflectivity and a random noise simulation and the weighting functions are integrated (cf. section 4.3). Next, we proceed to the computation of average $X_{CH_4}$ on 50 km resolution with the different averaging schemes (AVQ not simulated) and the correction algorithms presented in Section 4.4 and Table 5.

In order to estimate the bias, the computation of an average column integrated methane concentration $X_{CH_4}$ from the shot-by-shot volume mixing ratio profiles, $\mathrm{vmr}_{CH_4,i}(p)$, is needed and will be computed as $\overline{X_{CH_4}}^T$ in Eq. (9).

In order to assess the performance of averaging schemes and bias correction algorithms, the standard deviation and mean of the difference $\Delta X$ between the $\overline{X_{CH_4}}^{scheme}$ estimated from one of the studied averaging scheme and the target value $\overline{X_{CH_4}}^T$ must be computed over a set of $M$ simulations. The number of simulation $M$ has to be high enough to compute the residual bias (empirical mean of $\Delta X$) with sufficient accuracy. Let's denote $\sigma$ the standard deviation of the distribution of the variable





$\Delta X$, $S_M = \langle \Delta X \rangle$ the empirical mean over $M$ samples (i.e. the empirical estimate of the bias of the averaging scheme). To get an estimate of the expected value of $\Delta X$ with an accuracy of 0.1 ppb with 90 % confidence, it requires approximately $M = 300000$ samples, according to the central limit theorem.

The typical standard deviation can also be evaluated from the sample and is approximately 22 ppb for the typical case (mean

reflectivity of 0.1).

### 4.3    Simulation of on-line and off-line lidar signals and IWF

Once the scene parameters are defined on the 50 km averaging window and the atmosphere is modeled, the on-line and off-line signals must be simulated. We first have to compute the deterministic values of the signals without noise and simulate the random noise that affects them. The values of the signals are determined by the scene reflectivity (for both on-line and off-line

signals) and by the atmospheric transmission (on-line signals only). From the weighting functions, the methane field and the pressure field, we compute the reference DAOD, denoted $\delta_i^{\text{true}}$, as the numerator of Eq. (5). Then the transmission for each double-shot according to:

$$(\tau_i^{\text{true}})^2 = \exp(-2 \cdot \delta_i^{\text{true}}) . \tag{38}$$

From them, considering the reflectivity, we are able to determine the relative value of the on-line and off-line mean signals:

$$\mu_i^{\text{off}} = \rho_i , \tag{39}$$

$$\mu_i^{\text{on}} = \rho_i \cdot (\tau_i^{\text{true}})^2 , \tag{40}$$

where $i$ is the shot index, $\tau_i$ is the transmission and $\rho_i$ is the reflectivity. Note that any constant affecting both on-line and off-line signals can be disregarded here.

Then, Gaussian random noise has to be added to the values of the signals. It is computed from the SNR that depends on the

number of photon reaching the detector (i.e. $\mu_i^{\text{on,off}}$). Figure 9 shows the theoretical dependence of the SNR to the reflectivity according to instrument characteristics.

The $IWF_i$ are simply computed by integrating the $WF_{i,j}$ on all pressure layers as the denominator of Eq. (5).

### 4.4    Tested averaging algorithms and bias corrections

The simulation tested the three averaging schemes describes in section 3.2: AVX (Table 1, line 1), AVD (Table 1, line 2) and

AVS (Table 1, line 3). Table 5 details the computational steps used for averaging, statistical bias and geophysical bias evaluation for the three schemes. For the AVX and AVD schemes, as explained in section 3.3.1, signals with at least one negative signal must be discarded to compute the shot DAOD. However, since signals are averaged first for the AVS scheme, the probability that one of the mean signal is negative is extremely small. Thus, no negative signal discarding is needed for the AVS scheme.

Concerning statistical bias evaluation, a SNR estimation is needed. It is directly estimated from instrument parameters and on-line and off-line signal strength. Once the SNR is estimated, as described in section 3.3, there are two options to evaluate the




statistical bias either using the Taylor expansion approximation or the numerical integral of a truncated normal distribution. Contrary to AVS, where Taylor expansion and numerical integral make negligible difference, for AVX and AVD, it is better to use the numerical integral as it is more accurate, and this is what is done here.

Type 1 geophysical bias, that affects the AVX scheme, is already compensated by weighting the average $X_{CH_4}$ and the average

statistical bias by the IWF. The AVD scheme is not affected by geophysical biases. However, type 2 and type 3 geophysical biases affects the AVS scheme. Type 2 geophysical bias is evaluated by Eq. (35) using the first order $X_{CH_4}$ of Eq. (30). The type 3 geophysical bias is not evaluated and will not be corrected because a correlation between reflectivity and CH4 concentration is unlikely to occur. Indeed, on a sub 50 km scale the typical atmospheric transport should smear out the CH4 concentration very effectively over areas larger than the small scale reflectivity jumps even if this is not true for narrow valleys.

## 5   Results

### 5.1   Comparison of averaging schemes

#### 5.1.1   Bias of averaging schemes without bias correction

The first results presented here are respective biases of each averaging scheme without any bias correction. Figure 10 shows the bias on the average $X_{CH_4}$ on the three scenes (Toulouse, Millau and Chamonix) for the three averaging schemes that have

been studied: averaging of signals (AVS), averaging of DAOD and IWF separately (AVD) and averaging of $X_{CH_4}$ (AVX), without any correction. The bias due to measurement noise and due to geophysical variation appears on the results as it is not yet subtracted. For the AVS scheme we compare the results with the uniformly weighted average IWF and the average of the IWF weighted by the off-line signal strength: $w_i[Q^{off}]$. For the AVD scheme, a uniform weight is considered. And, for the AVX scheme, we compare the uniformly weighted average $X_{CH_4}$ (Eq. (23)) and the average $X_{CH_4}$ weighted by the IWF:

$w_i[IWF]$ (Eq. (24)). The mean reflectivity is set to the typical value of 0.1.

For the AVS scheme on Toulouse and Millau scenes where there are medium to high variations of geophysical quantities, the bias is contained in the ±1 ppb range. However, it is higher on the Chamonix scene where there are very high variations of geophysical parameters. As expected, the bias for the AVS scheme is mainly impacted by the variations of the geophysical parameters over the scene. Note that, on the Chamonix scene, the weighting of the average IWF by the off-line signal strength

reduces this bias.

On the contrary, the bias of the AVD and AVX schemes is not affected by the geophysical variations but is mainly driven by the measurement noise which essentially depend on the scene mean reflectivity. As shown in section 3.4.1, the AVD scheme with uniform weighting and the AVX scheme weighted by the integrated weighting function ($w_i[IWF]$ weights) show the same bias. Although the comparison between the AVX scheme weighted uniformly (light red on Figure 10) and the AVD

(green on Figure 10) shows that their biases are close when variations of surface pressure (main cause of variations of IWF) are low (Toulouse, Millau) but become significant when variations are higher (Chamonix).





Without any correction and for the typical reflectivity, the AVS scheme is less biased than the AVD and AVX schemes. However, as we have seen in previous sections, there are ways to estimate the biases and to correct them. The following section will show the results after estimation and correction of the bias induced by the measurement noise.

### 5.1.2    After correction of statistical bias

As explained in section 3.4, the random nature of the measurement associated with the non-linearity of the measurement equation implies that the estimation of the $X_{CH_4}$ is biased. The statistical bias corrections for AVS, AVD and AVX is based on an estimation of the on-line and off-line SNRs for the measured signals (cf. section 4.4 and Table 5). Figure 11 shows the residual biases, after subtraction of estimated statistical bias, for the three averaging schemes, with and without relevant weightings, on the three studied scenes and for the typical reflectivity of 0.1. The chosen estimation of the bias is done by
numerically computing the integral of the truncated gaussian distribution (section 3.3).

We see that the biases of the AVD and AVX schemes are significantly reduced (absolute value decrease by 85 to 90 %) on every scene. The residual bias is caused by the fact that the SNR are estimated from the noisy signals so that the estimation of the bias is not perfectly accurate. This implies that the signal outcomes from the lower part of the distribution leads to a high error on the estimated bias. This effect could be slightly compensated if instead of discarding all the negative or null signals
(extremely rare for a reflectivity value of 0.1 over 150), we discarded signals higher than a strictly positive threshold (e.g. 0.01, not shown). This would lead to a better correction and thus, a lower bias but at the cost of discarding more single shot observations.

For the AVS scheme, as the signals are averaged first, the equivalent SNR is very high ($SNR_{eq}^{off} \approx 190$ and $SNR_{eq}^{on} \approx 90$) on the scenes with a mean reflectivity of 0.1. Consequently, the bias due to the equivalent measurement noise is really low (about
0.1 ppb) and this bias correction has only a small effect on the residual bias.

Taking into account the correction of the bias induced by the measurement noise, the AVS scheme still present a lower bias on Toulouse and Millau scenes than the bias of AVX and AVD schemes. However, on the Chamonix scene, where the geophysical variations are very high, AVX and AVD schemes are less biased than AVS.

### 5.1.3    After correction of geophysical biases

The biases induced by the variation of the geophysical parameters (cf. section 3.4) does not affect the AVD scheme as the additive properties of DAOD and IWF are averaged separately. The variation of the IWF affects the bias of the AVX scheme and has already been corrected by introducing the $w_i[IWF]$ weights when directly averaging mixing ratios. The AVS scheme is the one most affected by the variations of the geophysical variations as seen in section 5.1.1.

Figure 12 shows the residual bias after the corrections of the statistical bias induced by the measurement noise and by the
variations of geophysical parameters (cf. section 4.4 and Table 5). We notice that the residual bias for the AVS scheme is





considerably reduced when the average weighting function is weighted by the off-line signal strength. Furthermore, the iterative estimation of the bias converges at the first iteration of Eq. (26) to Eq. (31).

Once geophysical biases are subtracted, the three scenes present a low bias. The mean residual bias on the three scenes for the AVD and AVX schemes is approximately $-2.1 \pm 0.1$ ppb, whereas for the AVS scheme, it is approximately $-0.09 \pm$

$0.09$ ppb. After all corrections, even on highly structured scenes, AVS is the least biased scheme of the three studied schemes. When the average IWF is consistently weighted with the $w_i[Q^{\text{off}}]$ weights, the geophysical induced bias is almost completely removed.

### 5.2 Impact of the mean reflectivity on the residual bias

All results presented above are computed for scenes with a mean reflectivity of 0.1 which roughly correspond to a vegetation

cover. For the purpose of choosing the least biased algorithm to compute average $X_{CH_4}$, it is interesting to test the robustness to reflectivity. Indeed, reflectivity is the main driver for the expected value of SNR: low reflectivity scenes lead to lower SNR and consequently higher bias. Table 6 and Table 7 show the residual bias comparing four different reflectivity values: 0.1 (vegetation), 0.05 (mixed sea/vegetation), 0.025 (sea/ocean) and 0.016 (ice/snow). Table 6 gives the residual bias for the AVD scheme (the AVX scheme present similar results) and Table 7 shows the residual bias for the AVS scheme where the average

IWF is weighted by the off-line signal strength ($w_i[Q^{\text{off}}]$ weights) and corrected of the bias due to geophysical variations from shot-to-shot. For both tables, Taylor expansion bias correction (Eq. (15)) or numerical computation of the expectation (so-called integral truncated normal distribution Eq. (16) and Eq. (13)) are compared.

First, as seen in Table 6 (AVD scheme), the Taylor bias correction does not succeed in quantifying the bias on any of the four mean reflectivity values. The uncertainties are too high and prevent quantitative analysis of the results. This is due to the fact

that there are some signals that are really close to zero and for which the SNR is underestimated and thus the bias (and standard deviation) overestimated. This could be mitigated by the choice of a higher threshold of usable signal before the computation of the DAOD (not shown). The results when using the integral bias correction on AVD are more physical. However, they also show an over estimation of the bias especially for low reflectivity values. In every case for the AVD scheme, the bias threshold is exceeded.

Table 7 gives the results of robustness of the AVS scheme to decreasing reflectivity. Unlike the AVD scheme, the AVS scheme, when all corrections are made, presents satisfying results for all reflectivity values and every scene, the biases remain contained into the threshold interval of $\pm 1$ ppb. The effect of the decreasing reflectivity has a very small impact on the residual bias.

To summarize, the best algorithm to limit the bias for MERLIN processing algorithms is clearly the AVS scheme with an average IWF weighted by the off-line signal strength and both correction of the geophysical bias and the bias induced by the

measurement noise (either Taylor or integral bias correction). On every scene and for all expected reflectivity values, this algorithm is compliant with the averaging bias specifications of the MERLIN mission. Note that this conclusion holds in the



case where all the 150 shots are considered; in the case of a partially cloudy window where only a subsample of clear sky shots are averaged, AVS will still be the best averaging scheme, but the performance will be decreased.

## 6   Conclusions

The French-German space-borne IPDA lidar mission MERLIN will measure the average integrated column dry-air mixing
ratio of methane ($X_{CH_4}$) on a 50 km scale. The processing algorithms must limit both the relative random error (RRE) and the relative systematic error (RSE) on the $X_{CH_4}$. As the IPDA technique relates the signal measurements to the $X_{CH_4}$ by a non-linear equation, a simple and naive averaging can lead to high biases.

Three averaging schemes have been studied: averaging of $X_{CH_4}$ (AVX), averaging of DAOD (AVD) and averaging of signals (AVS). For these averaging schemes, possible sources of bias can either be the measurement noise or the variation of the
geophysical parameters on the averaging scene or both.

The three schemes are sensitive to the bias induced by the measurement noise even if AVS is far less impacted for the typical reflectivity. This bias can be corrected by a formula introducing the estimated SNR on the measured signals if the SNR is high enough. The bias due to the variation of geophysical parameters does not affect the AVD scheme because it directly averages the desired additive quantities. On the contrary, the AVX scheme must average the concentration weighted by the integrated
weighting function (IWF) in order to average a molecule number instead of averaging concentrations. The third scheme AVS measures the average $X_{CH_4}$ weighted by the off-line signal strength which means that more weight to the measurements with high SNR is given when averaging. The bias of this scheme is sensitive to the variation of geophysical parameters (surface pressure and surface reflectivity). This bias is corrected using an iterative process with the uncorrected $X_{CH_4}$ as first-guess.

These averaging schemes and their bias corrections have been tested on scenes modeled from real satellite data in terms of
altitude, surface pressure, weighting functions and relative variations of reflectivity. The three scenes present interesting characteristics as they show different geophysical variations that could impact averaging biases. Besides, the signals and random noise are simulated form geophysical parameters and instrument parameters.

The simulation shows that the lowest biases are obtained for the AVS scheme using appropriate bias corrections and averaging weights. Furthermore, this scheme is robust to low reflectivity values unlike the AVX and AVD schemes which are highly
sensitive to the accuracy of the SNR estimation. The best scheme, AVS, is compliant with the allocated averaging bias requirements (RSE) of 0.07% (1 ppb for a $X_{CH_4}$ of 1780 ppb) for the whole range of expected reflectivity values (from 0.1 down to 0.016).

The continuation of this study could evaluate the sensitivity of a poor (unprecise or biased) estimation of the SNR on the estimation of the bias due to measurement noise for low reflectivity values. Furthermore, the use of the LIDAR simulator and
processor suites, currently in development at the LMD, could be beneficial to the evaluation of the biases, and more specifically of averaging biases, on a wider scale (many scenes, atmosphere types…)



## Appendix A

The averaging of quotients estimates the average of the shot-by-shot two-way transmissions $\tau_i^2$. Due to the measurement noise, we will suppose that, for the shot $i$, the on-line and off-line measured signals, resp. $Q_i^{on}$ and $Q_i^{off}$, are outcomes of normal distributions with mean values resp. denoted $\mu_i^{on}$ and $\mu_i^{off}$, and standard deviation resp. denoted $\sigma_i^{on}$ and $\sigma_i^{off}$. The $X_{CH_4}$

computed from averaging quotients can be defined as:

$$\overline{X_{CH_4}}^{avq} = \frac{-\frac{1}{2} \cdot \ln\langle\tau^2\rangle}{\langle IWF\rangle}, \tag{A1}$$

with the transmission defined as:

$$\langle\tau^2\rangle = \frac{1}{N}\sum_{i=1}^{N}\frac{Q_i^{on}}{Q_i^{off}}. \tag{A2}$$

If we define the standardized signals corresponding to $Q_i^{on}$ and $Q_i^{off}$ as $X_i^{on}$ and $X_i^{off}$, the average transmission can be written:

$$\langle\tau^2\rangle = \frac{1}{N}\sum_{i=1}^{N}\frac{\mu_i^{on} + \sigma_i^{on}\cdot X_i^{on}}{\mu_i^{off} + \sigma_i^{off}\cdot X_i^{off}}. \tag{A3}$$

Then we can further separate the random part due to measurement noise and the deterministic part due to varying geophysical parameters as follows:

$$\langle\tau^2\rangle = \frac{1}{N}\sum_{i=1}^{N}\left[\frac{\mu_i^{on}}{\mu_i^{off}}\left(1 + \frac{\sigma_i^{on}}{\mu_i^{on}}X_i^{on}\right)\left(1 + \frac{\sigma_i^{off}}{\mu_i^{off}}X_i^{off}\right)^{-1}\right], \tag{A4}$$

$$\langle\tau^2\rangle = e^{-2\langle\delta^{true}\rangle}\cdot\frac{1}{N}\sum_{i=1}^{N}\left[e^{-2\Delta\delta_i}\left(1 + \frac{\sigma_i^{on}}{\mu_i^{on}}X_i^{on}\right)\left(1 + \frac{\sigma_i^{off}}{\mu_i^{off}}X_i^{off}\right)^{-1}\right], \tag{A5}$$

where $\langle\delta^{true}\rangle$ is the average DAOD computed from noiseless mean signals and $\Delta\delta_i$ the difference to the shot-by-shot DAOD. Then we can deduce the error of AVQ scheme as follows:

$$\overline{X_{CH_4}}^{avq} = X_{CH_4}^{true} - \frac{1}{2\langle IWF\rangle}\cdot\ln\left(\frac{1}{N}\sum_{i=1}^{N}\left[e^{-2\Delta\delta_i}\left(1 + \frac{\sigma_i^{on}}{\mu_i^{on}}X_i^{on}\right)\left(1 + \frac{\sigma_i^{off}}{\mu_i^{off}}X_i^{off}\right)^{-1}\right]\right). \tag{A6}$$

The corresponding bias is the expected value of the error term:

$$Bias\left(\overline{X_{CH_4}}^{avq}\right) = -\frac{1}{2\langle IWF\rangle}\cdot E\left[\ln\left(\frac{1}{N}\sum_{i=1}^{N}\left[e^{-2\Delta\delta_i}\left(1 + \frac{\sigma_i^{on}}{\mu_i^{on}}X_i^{on}\right)\left(1 + \frac{\sigma_i^{off}}{\mu_i^{off}}X_i^{off}\right)^{-1}\right]\right)\right]. \tag{A7}$$

In equation (A5), the empirical average transmission is decomposed into two factors. The first is the transmission corresponding to the average DAOD from noiseless signals. The second factor is the average of multiplicative errors that are the deterministic error from geophysical variations on the averaging scene and the random factors due to the presence of measurement noise. As shown in Eq. (A7), the error sources are mixed into the non-linear function which makes them difficult to evaluate. It is possible to derive a suitable bias correction based on Eq. (A7) for AVQ but in the end it is not expected to be

better than the other ones.

**Author contribution.** Clémence Pierangelo and Yoann Tellier designed the simulation algorithms while Yoann Tellier handled its implementation with the support of Fabien Gibert. Martin Wirth developed theoretical aspects such as averaging



scheme definition, or iterative geophysical bias correction and supported the whole work. Fabien Marnas provided the data set used for the real scene model and the interpolated surface pressures. Yoann Tellier prepared the manuscript with contributions from all authors.

**Data availability.** Data set will not be made available as some data are disclosable to the whole community.

**Competing interests.** The authors declare that they have no conflict of interest.

**Acknowledgements.** This work was funded by CNES as part of the CNES/DLR project MERLIN. We thank Frédéric Chevallier (LSCE) for the kind support he provided to this work. Authors would also like to thank the following LMD collaborators working on MERLIN project (in alphabetical order): Raymond Armante, Vincent Cassé, Olivier Chomette, Cyril Crevoisier, Thibault Delahaye, Dimitri Edouart and Frédéric Nahan.

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





| Averaging Scheme | Acronym | Definition | Bias Characteristics |
|---|---|---|---|
| Averaging of columns of $X_{CH_4}$ | AVX | $\overline{X_{CH_4}}^{avx} = \langle \frac{\delta}{IWF} \rangle_{w[IWF]}$ | – Statistical bias due to measurement noise on every shot<br>– Type 1 geophysical bias from averaging concentrations instead of molecular content |
| Averaging of columns of DAOD and IWF | AVD | $\overline{X_{CH_4}}^{avd} = \frac{\langle \delta \rangle}{\langle IWF \rangle}$ | – Statistical bias due to measurement noise on every shot |
| Averaging of signals | AVS | $\overline{X_{CH_4}}^{avs} = \frac{\frac{1}{2} \cdot \ln \left( \frac{\langle Q^{off} \rangle}{\langle Q^{on} \rangle} \right)}{\langle IWF \rangle_{w[Q^{off}]}}$ | – Statistical bias due to measurement noise of the resulting signals on the averaging window<br>– Type 2 geophysical bias due to linearization of the DAOD variations and correlation between DAOD and reflectivity variations<br>– Type 3 geophysical bias due to the higher sensitivity to measurements with high off-line signal strength |
| Averaging of quotients (not detailed in this paper due to bad performances) | AVQ | $\overline{X_{CH_4}}^{avq} = \frac{\frac{1}{2} \cdot \ln \langle \frac{Q^{off}}{Q^{on}} \rangle}{\langle IWF \rangle}$ | – Statistical bias due to measurement noise mixed with geophysical biases into the non-linear equation (cf. Appendix **Erreur ! Source du renvoi introuvable.**) |

**Table 1: Averaging schemes and characteristics of their biases**





| Reflectivity value | 0.093 | 0.077 | 0.062 | 0.53 | 0.025 |
|---|---|---|---|---|---|
| Off-line SNR | 15.1 | 13.1 | 10.9 | 9.5 | 4.8 |
| On-line SNR | 6.1 | 5.2 | 4.2 | 3.6 | 1.8 |
| Error made by Taylor expansion (Eq. (15)) | -1 ppb | -2 ppb | -5 ppb | -10 ppb | +50 ppb |

**Table 2: Error on the statistical bias estimation by using the Taylor expansion instead of using truncated normal distribution (cf. Figure 4)**





| Geophysical parameter | Origin | Original grid resolution | Interpolated grid resolution |
|---|---|---|---|
| **Coordinates (lat., lon.)** | SPOT-5 | 20 m | 90 m |
| **Relative reflectivity** | | | (surface pressure is corrected to take into account SRTM small scale variations of topography) |
| **Altitude** | SRTM | 90 m | |
| **Surface pressure** | ECMWF | ~ 10 km | |

**Table 3: Data sets resolution characteristics**



| | Toulouse | Millau | Chamonix |
|---|---|---|---|
| **Latitude range** | 43.56° N–43.93° N | 43.56° N–43.93° N | 45.75° N–46.12° N] |
| **Longitude** | 1.62° E | 3.06° E | 7.22° E |
| **Altitude range (m)** | 108–321 | 359–902 | 473–2967 |
| **Altitude mean (m)** | 223.1 | 697.4 | 1753.7 |
| **Altitude standard deviation (m)** | 57.5 | 141.8 | 711.1 |
| **Surface pressure range (hPa)** | 980.2–1000.9 | 922.7–973.9 | 748.2–965.0 |
| **Surface pressure mean (hPa)** | 988.7 | 940.9 | 837.3 |
| **Surface pressure standard deviation (hPa)** | 5.7 | 12.2 | 64.5 |
| **Relative reflectivity range** | 0.68–1.65 | 0.49–1.50 | 0.35–1.61 |
| **Relative reflectivity standard deviation** | 0.16 | 0.24 | 0.27 |

**Table 4: Characteristics of the data used for the simulation**



|  | **AVX scheme** | **AVD scheme** | **AVS scheme** |
|---|---|---|---|
| **Averaging scheme** | – Discard negative signals<br>– Table 1, line 1 | – Discard negative signals<br>– Table 1, line 2 | – Table 1, line 3 |
| **Statistical bias evaluation** | – Discard negative signals<br>– SNR estimation<br>– Bias evaluation:<br>  Option 1: Eq. (33) and (15)<br>  Option 2: Eq. (33), (16) and (13) | – Discard negative signals<br>– SNR estimation<br>– Bias evaluation:<br>  Option 1: Eq. (32) and (15)<br>  Option 2: Eq. (32), (16) and (13) | – SNR estimation<br>– Equivalent window SNR by Eq. (21)<br>– Bias evaluation:<br>  Option 1: Eq. (34) and (15)<br>  Option 2: Eq. (34), (16) and (13) |
| **Geophysical bias evaluation** | – None (Type 1 geophysical bias built-in $w[IWF]$ weights of Table 1, line 1 and Eq. (33)) | – None | – Type 2 geophysical bias of Eq. (35)<br>– Type 3 geophysical bias of Eq. (36) not estimated |

**Table 5: Computational details about averaging schemes and bias evaluation**





| | Taylor Bias Correction (not usable) | | | | Integral Bias Correction | | | |
|---|---|---|---|---|---|---|---|---|
| **Reflectivity** | 0.1 | 0.05 | 0.025 | 0.016 | 0.1 | 0.05 | 0.025 | 0.016 |
| **Off-line SNR** | 16.1 | 9.0 | 4.8 | 3.2 | 16.1 | 9.0 | 4.8 | 3.2 |
| **On-line SNR** | 6.5 | 3.4 | 1.8 | 1.1 | 6.5 | 3.4 | 1.8 | 1.1 |
| **Toulouse (ppb)** | -6.70e-1 | -9.02e4 | -9.70e10 | -5.11e9 | -1.79 | -4.28 | 207 | 416 |
| **Millau (ppb)** | -1.52e2 | -8.94e5 | -6.27e7 | -4.62e8 | -2.43 | 9.69 | 204 | 442 |
| **Chamonix (ppb)** | -1.19e1 | -1.59e6 | -4.35e9 | -2.44e8 | -2.05 | 8.24 | 197 | 498 |
| **Uncertainty (ppb)** | ±6.7e1 | ±8.8e5 | ±5.5e10 | ±2.0e9 | ±0.10 | ±0.24 | ±0.61 | ±0.89 |

**Table 6: Resulting bias (in ppb) for AVD scheme after noise induced bias correction**





| | Taylor Bias Correction | | | | Integral Bias Correction | | | |
|---|---|---|---|---|---|---|---|---|
| **Reflectivity** | 0.1 | 0.05 | 0.025 | 0.016 | 0.1 | 0.05 | 0.025 | 0.016 |
| **Off-line SNR** | 16.1 | 9.0 | 4.8 | 3.2 | 16.1 | 9.0 | 4.8 | 3.2 |
| **On-line SNR** | 6.5 | 3.4 | 1.8 | 1.1 | 6.5 | 3.4 | 1.8 | 1.1 |
| **Toulouse (ppb)** | -0.01 | -0.02 | -0.03 | -0.05 | -0.03 | -0.07 | -0.03 | -0.08 |
| **Millau (ppb)** | -0.03 | -0.03 | -0.03 | -0.05 | -0.04 | -0.08 | -0.04 | -0.08 |
| **Chamonix (ppb)** | -0.51 | -0.51 | -0.50 | -0.48 | -0.53 | -0.57 | -0.50 | -0.49 |
| **Uncertainty (ppb)** | ±0.09 | ±0.17 | ±0.33 | ±0.51 | ±0.09 | ±0.17 | ±0.33 | ±0.51 |

Table 7: Resulting bias (in ppb) for AVS scheme after noise induced bias correction and geophysical induced bias correction



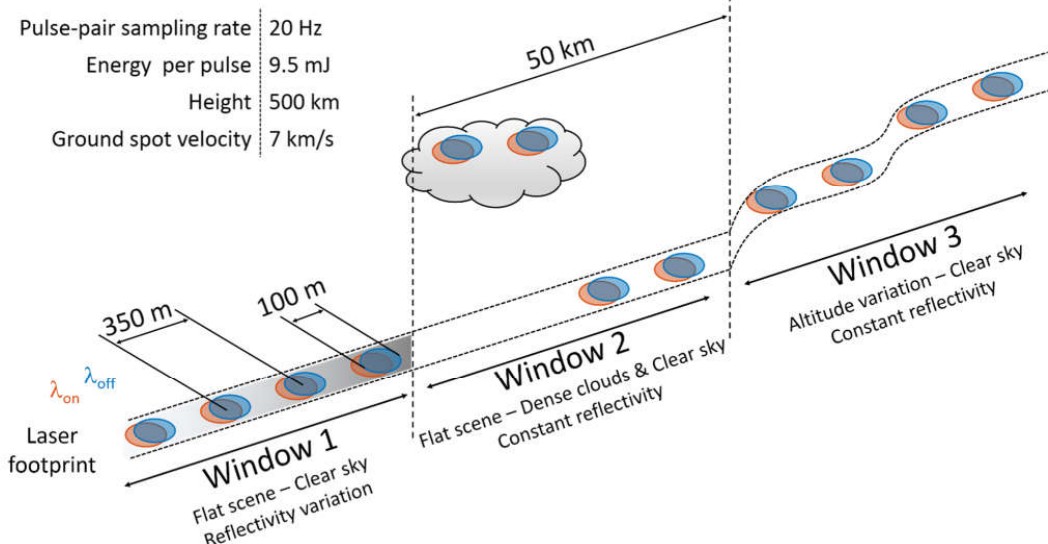

**Figure 1: Principle schematics of the MERLIN IPDA lidar measurement. The lidar emits two laser beams with slightly different wavelengths ($\lambda_{on}$ and $\lambda_{off}$). Every measurement corresponds to the small fraction of the two laser beams – called on-line and off-line signals – that are reflected by a "hard" target (Earth's surface, top of dense clouds) to the satellite receiver telescope. For clarity, the three averaging windows are represented with four measurements instead of 150. On every averaging window, geophysical parameters altitude (or scattering surface elevation when there are clouds) or reflectivity vary.**



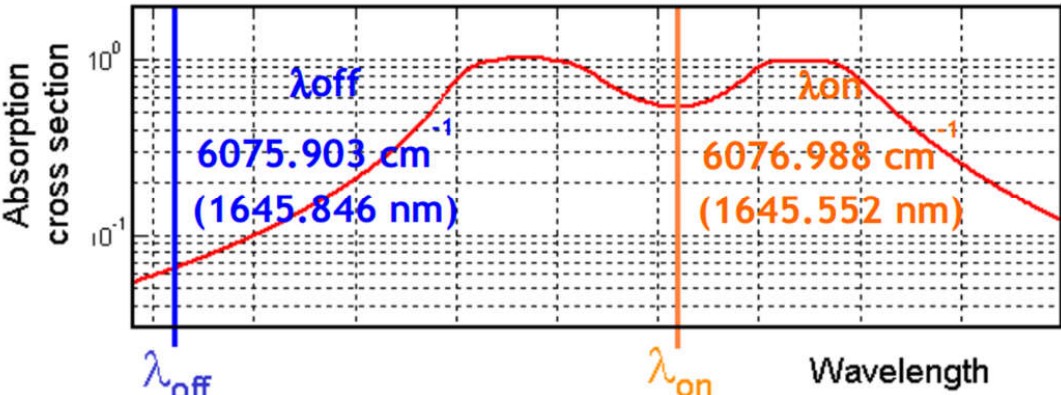

**Figure 2: Laser frequency positioning of the on-line and off-line laser beams. The on-line frequency is positioned in the through of one of the methane absorption line multiplets. The off-line frequency is positioned so that the methane absorption is negligible.**





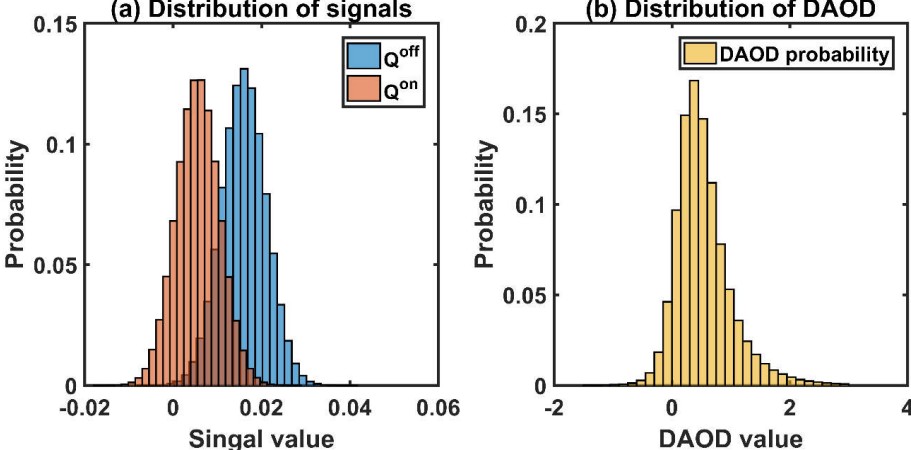

**Figure 3:** Effect of the Non-linearity on the DAOD Distribution for a low reflectivity (0.016 ice/snow cover). Panel (a) shows that the on-line and off-line signals are normally distributed. A significant part of on-line signals (orange) is negative which makes the corresponding double-shots unusable (undefined logarithm). Panel (b) shows that the DAODs corresponding to the usable signals are not normally distributed but skewed. The true DAOD is 0.53 whereas the mean of the distribution is about 0.54 which leads to a bias on the $X_{CH_4}$ of approximately +34 ppb here.





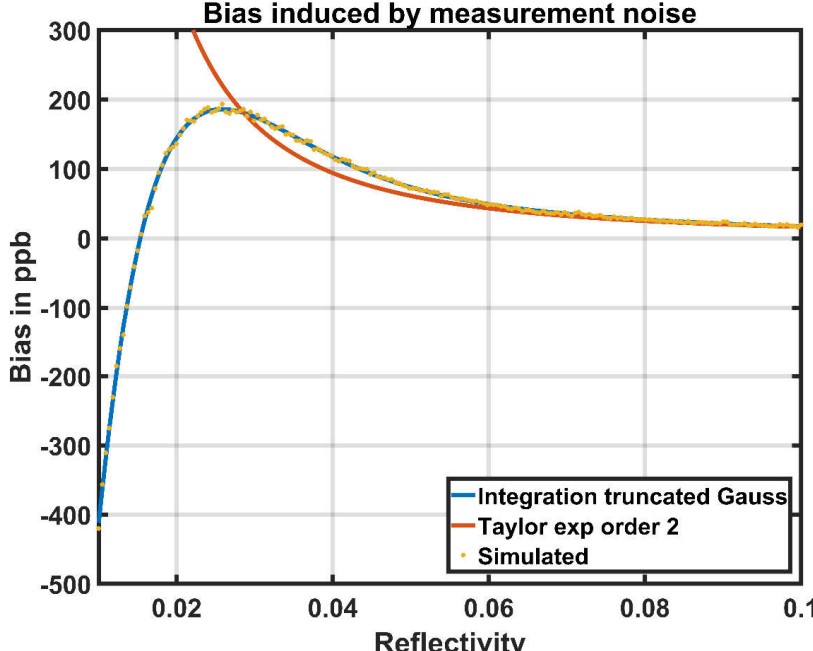

**Figure 4: Statistical bias induced by measurement noise.** The on-line and off-line SNR drive the value of the statistical bias. The **blue** line is derived from the integration of the truncated normal distribution (Eq. (16) and (13)). The **orange** line is the Taylor development (Eq. (15)) only valid when reflectivity is high enough (i.e. high SNR). The expected bias computed from a simple Monte-

5  Carlo simulation (**Yellow** dots) shows that the integration approach is the most accurate. For reflectivity values of 0.1 (vegetation cover), integration (**blue**) and Taylor development (**orange**) differs of about 1 ppb (cf. Table 2 for some values).

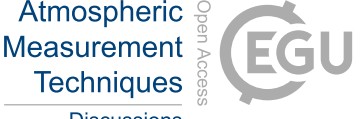



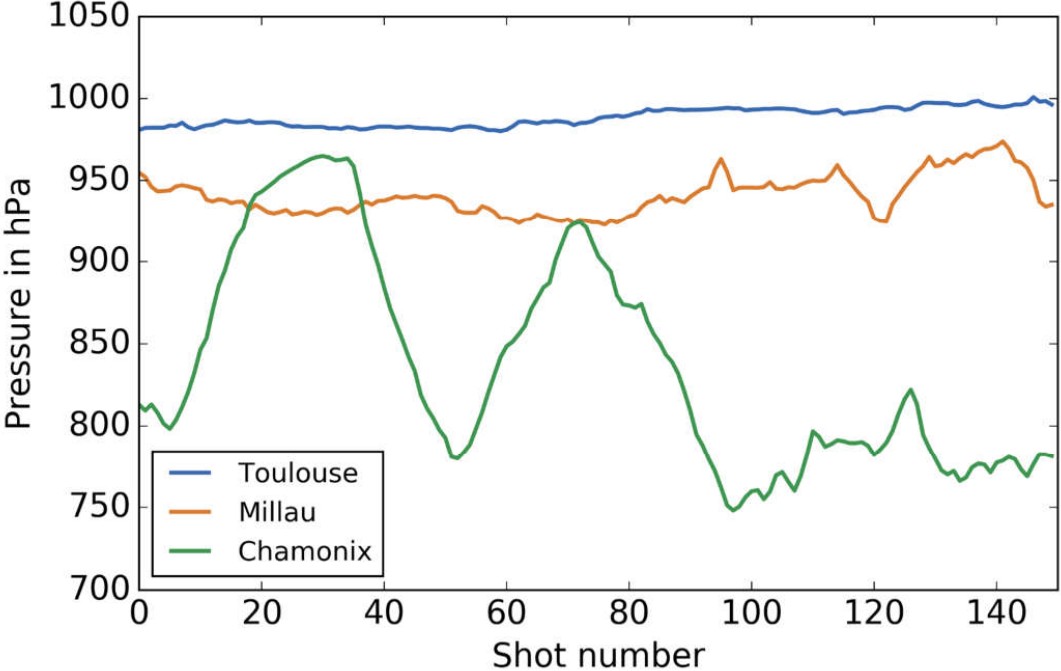

**Figure 5: Surface pressure of the three scenes from the data sets. Toulouse (resp. Millau, Chamonix) presents medium (resp. high, very high) variability (cf. Table 4).**




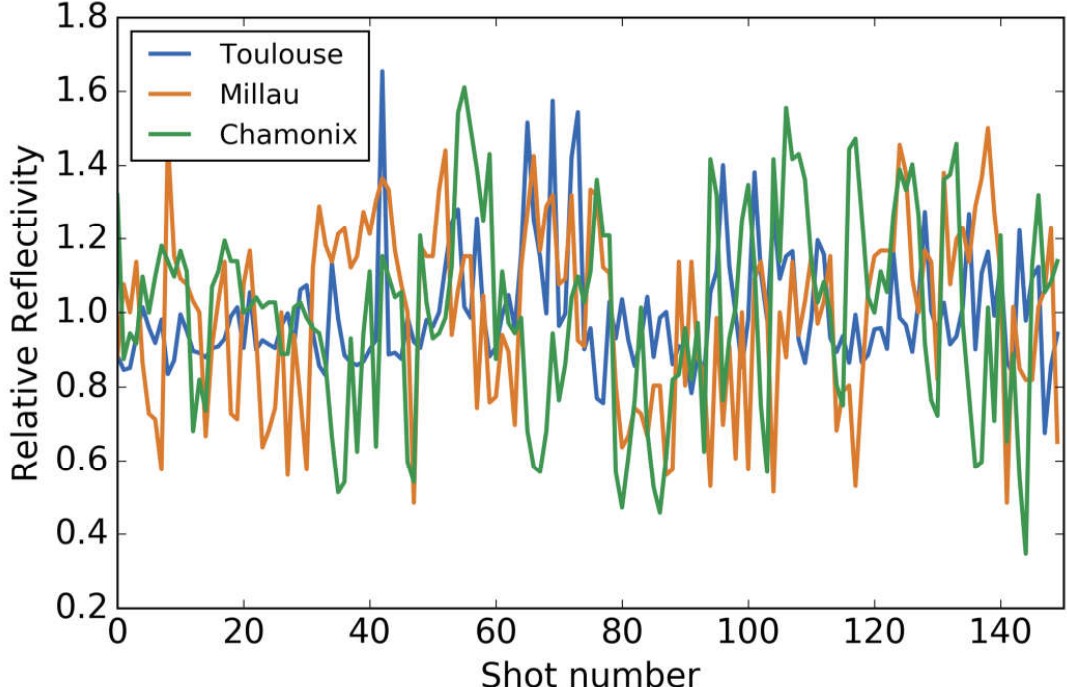

**Figure 6: Relative variations of reflectivity of the three scenes from the data set. Toulouse (resp. Millau, Chamonix) presents medium (resp. high, high) variability (cf. Table 4).**



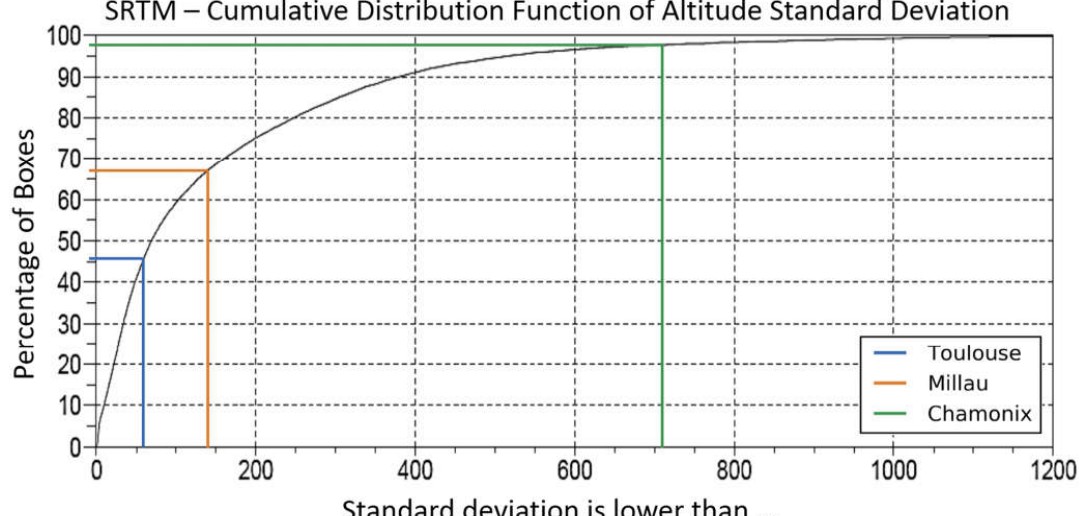

**Figure 7: Global cumulative distribution of standard deviation of altitude obtained on SRTM. 46% (resp. 67%, resp. 97%) of SRTM boxes present a lower standard deviation than Toulouse scene (resp. Millau scene, resp. Chamonix scene). The three scenes are representative of medium, high and very high variations of altitude.**


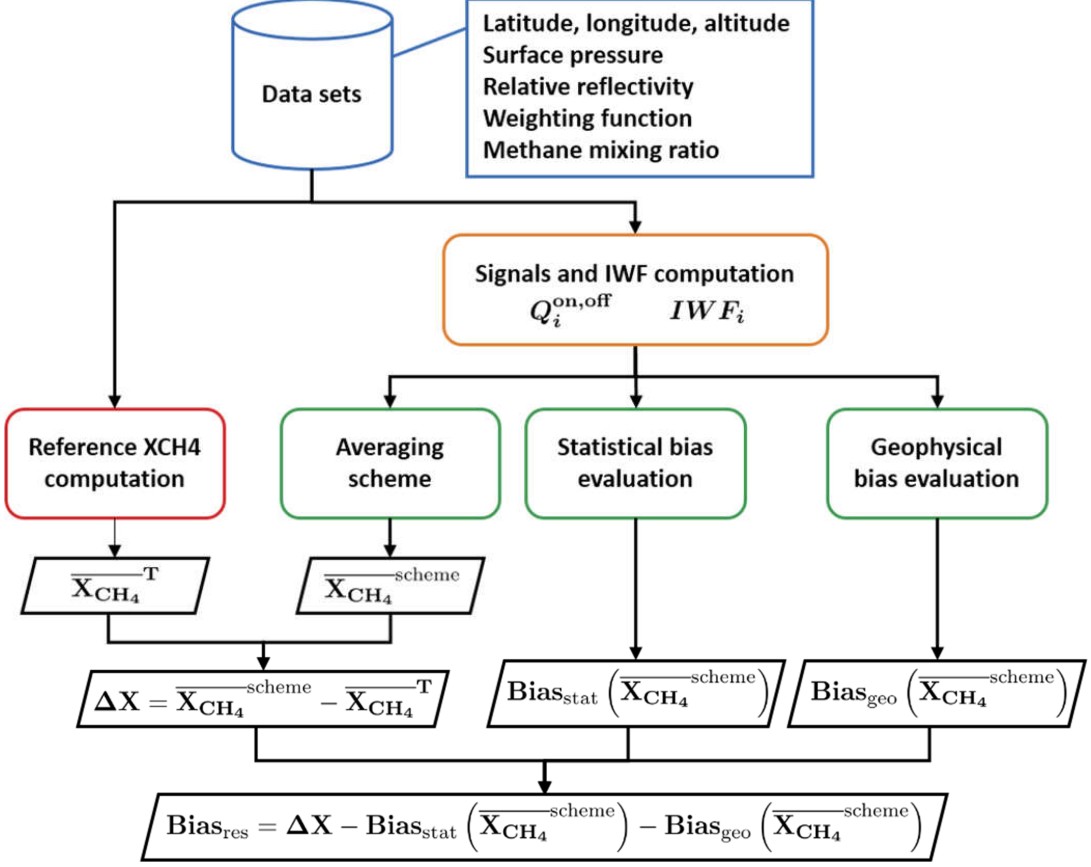

**Figure 8: Global description of the simulation. Data sets (blue) are described in section 4.1. Signals and IWF computation (orange) is described in section 4.3. Averaging strategies performed and their related bias corrections (green) are described in section 4.4 and Table 4. Target $X_{CH_4}$ computation (red) is described in section 3.1. $\Delta X$ is the scheme bias which is the difference between scheme and target $X_{CH_4}$ and $Bias_{res}$ is the residual bias when evaluated biases have been subtracted from the scheme bias.**





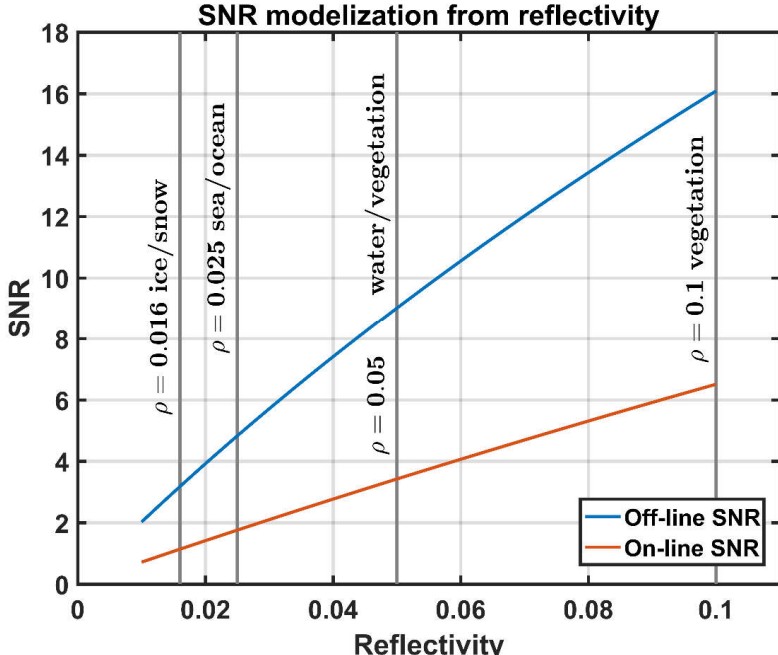

**Figure 9: On-line and off-line SNR computed from reflectivity according to instrument characteristics**





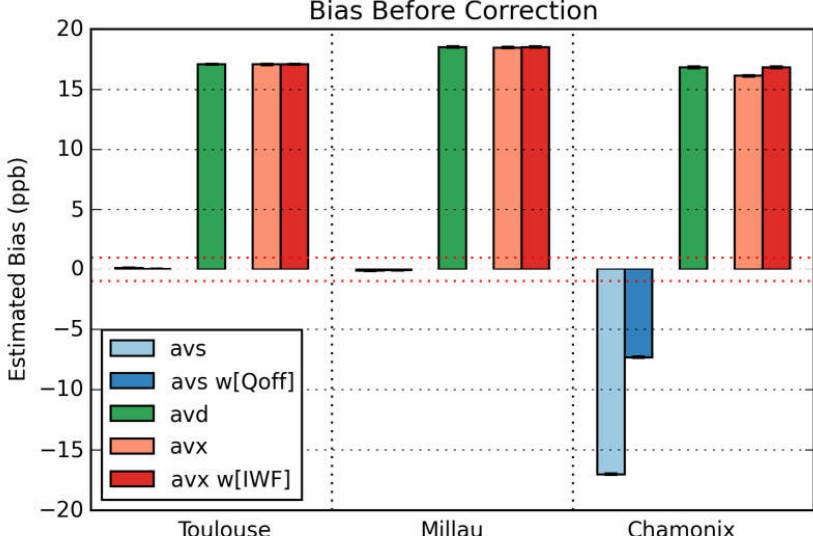

Figure 10: Bias before correction for the three studied averaging scheme (red dotted lines: targeted bias ±1 ppb).





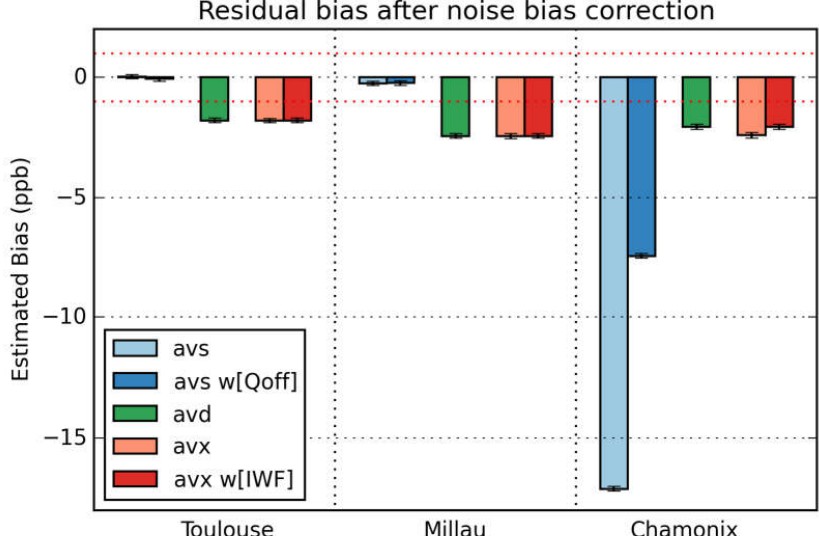

**Figure 11: Residual bias after statistical bias correction for the three studied averaging scheme (red dotted lines: targeted bias ±1 ppb).**





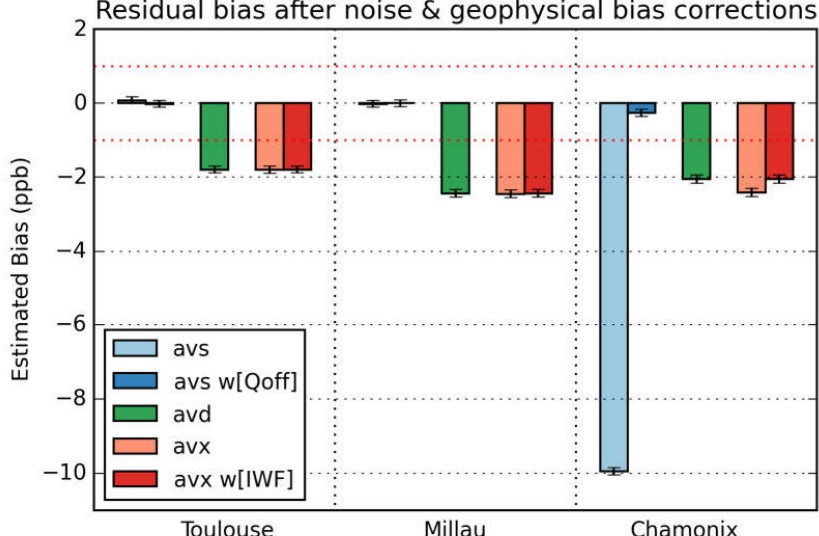

**Figure 12: Residual bias after noise induced bias and geophysical variation induced bias corrections for the three studied averaging scheme (red dotted lines: targeted bias ±1 ppb).**