# Peer review of "Averaging Bias Correction for the Future Space-borne Methane IPDA Lidar Mission MERLIN"

_Atmospheric Measurement Techniques, 2018_

## Referee Comment (RC1) · Anonymous Referee #1 · 17 Apr 2018

Overall Comments: The paper is in principle very interesting and tackles an important issue: how to best average IPDA lidar signals to reduce biases. The authors present several averaging schemes and evaluate the merits of each technique for the MERLIN mission. They take great care to use real data sets for latitude, longitude, altitude, surface pressure, relative reflectivity and meteorological data. However, their lidar analysis does not seem to be based on any real MERLIN parameters (e.g. laser power, detector noise, shot noise, etc). Instead, the authors seem to treat the lidar signals as purely mathematical quantities that can even take negative values. No real SNR and subsequent error or bias analysis based on the real MERLIN instrument is presented. The analyses by Ehret and Kimmel treat the errors correctly and relate them to physical quantities. Those papers could serve as a good guide. A key issue in their analysis

is "measurement noise". In fact, the statistical bias induced by measurement noise is central to the analysis. However, that measurement noise is never defined and its characteristics are never described. Is it shot noise? Detector noise? Or some other "measurement" noise? Is it always purely a Gaussian, or a Poisson or some other distribution? What is the source of the "measurement noise"? These assumptions and questions are central to their analysis but are not well defined.

In summary, it is difficult to agree with their conclusions given their purely mathematical approach. I would suggest they revise the paper and apply their analysis starting with the error analysis already presented by Ehret and Kimmel.

Abstract: This article discusses how to process horizontal averaging in order to avoid the bias caused by the non-linearity of the measurement equation with measurements affected by random noise and horizontal geophysical variability. This sentence is a bit confusing. Did the authors mean: This article discusses how to process this horizontal averaging in order to avoid the bias caused by the non-linearity of the measurement equation and measurements affected by random noise and horizontal geophysical variability?

To be precise: 0.07% of 1780 ppb is 1.25 ppb (see similar comment below)

Introduction: This technique relies on the Differential Absorption Lidar (DIAL) measurements of a space-borne laser. This sentence is somewhat redundant considering the previous sentence: "...based on an Integrated Path Differential Absorption (IPDA) lidar. If the authors wish to make a distinction between IPDA and DIAL or further explain the technique then I suggest they should do so.

Figure 2 is mentioned before Figure 1

I believe operational analyzes should be operational analyses (plural of the noun). Analyze (British spelling) is the verb.

ASCENDS mission (NASA carbon dioxide IPDA lidar mission) – the appropriate ref-

erence listed in the references is needed. In addition, I believe it is best to define acronyms fully the first time they are used.

The former bias being negligible compared to the latter. If I interpreted this statement correctly then the authors argue that: "...biases caused by real system drift..." are negligible compared to biases that are caused by the non-linearity of the IPDA lidar measurement equation. Is this justified by any experimental (or theoretical) evidence of the biases caused by real (MERLIN) system drift? I do not see how this statement is supported by any evidence. In fact, the reference cited earlier in the paragraph (Werle 1993) actually discussed biases caused by real system drifts.

1% of 1780ppb is 17.8 ppb and 0.2% is 3.6 ppb. I understand the authors round up or down but they just need to be consistent.

Given the statement: the single shot on-line and off-line random error is reduced by a factor of sqrt(150)~12, I don't quite understand the example the authors gave: For instance, for the typical reflectivity (0.1), the on-line and off-line signal to noise ratios are of the order of 7 and 16 respectively (resp.) and the equivalent SNRs for the averaged signals are resp. 78 and 192. But sqrt(150)*7 =86 and sqrt(150)*16=196 not 78 and 192.

This process greatly decrease the RRE should be: This process greatly decreases the RRE

Overview of IPDA equations and the MERLIN processing chain vmr looks like vnr after eq. (3) in my pdf copy

Averaging schemes and bias correction: a theoretical approach The expected value of an random variable should be: The expected value of a random variable

vmr looks like vnr in my pdf copy

...then derived from the pressure at every levels should be:...then derived from the pressure at every level.

I really do not understand the argument: Figure 3 illustrates the statistical bias, when on-line and off-line signals follow normal distributions. It highlights that, in this case, the DAOD derived from these signals is no longer normally distributed but skewed. First, I do not understand how Qon can be negative. Then even though I agree that the ratio of two normal distributions can produce a skewed distribution but that does not equate with bias. The DAOD could be skewed but have zero bias.

This sentence needs to be rewritten. It is confusing and not at all clear what the authors are trying to say: Consequently, this scheme is more sensitive to the less noisy measurements which, on the one hand, implies that the variance of average quantities is lower but, on the other hand, a correlation between methane concentration and reflectivity implies a bias.

Link missing: (cf. Appendix Erreur ! Source du renvoi introuvable.).

This scheme gives very bad performances should be: This scheme gives very bad performance.

The fourth scheme AVQ is dismissed because it "gives very bad performance". It would have been much better to show the bad performance otherwise we just have to trust the authors.

The total noise contributions affecting off-line and on-line signals are statistically independent. Are they statistically independent? How do we know that? Can they be correlated if they have the same source? Does that change the results?

Furthermore, due to the relatively high number of photons in a single pulse, we can assume that these random variables are normally distributed around a mean value. There was no mention about the actual number of photons in a single pulse prior to this statement so we do not actually know how many photons are in the signals, Q. If we are photon (shot) noise limited the distribution should be Poisson not normal. Also, in the cases where the SNR is low (i.e. "the relatively high number of photons" is low)

does the randomness assumption still hold?

I fail to understand how either one or both lidar signals can take negative values. What does it physically mean for a lidar return signal to be negative? If the signal is completely absorbed or is very low (because of clouds or low reflectivity) then it can be zero but not negative.

I do not agree with the statement: This can actually happen as $Q_{on,off}$ correspond to the photon count from which the background has been subtracted. I do not agree that this explains "negative" signals! It means there is no signal and all we have is background noise (could be solar background or detector noise). That does not make the lidar signal negative!

Methodology to test averaging algorithms and their bias corrections How are the on-line and off-line signals computed purely from surface reflectivity and a random noise simulation? What assumptions were made for laser power, field of view, detector NEP etc etc.? Figure 8 fails to illuminate how $Q_{on}$ and $Q_{off}$ are computed.

Figure 9 shows the theoretical dependence of the SNR to the reflectivity according to instrument characteristics (assumptions) but those characteristics are never listed.

I may be missing the point here but I fail to understand how a higher threshold of usable signal before the computation of the DAOD could mitigate the fact that the Taylor bias correction does not succeed in quantifying the bias on any of the four mean reflectivity values. That is equivalent of excluding low SNR cases, which may be valid for data quality control, but does not mitigate the fact that the approach does not succeed in those cases.

Figures Figure 1: On every averaging window, geophysical parameters altitude (or scattering surface elevation when there are clouds) or reflectivity vary. Should be: On every averaging window, geophysical parameters such as altitude (or scattering surface elevation when there are clouds) and reflectivity vary.

Figure 3: A significant part of on-line signals (orange) is negative – I do not understand how a signal can be negative. What part of Eq.(1) is negative for Qon? I could be missing something but it is not clear.

Table 1: Link error: Statistical bias due to measurement noise mixed with geophysical biases into the non-linear equation (cf. Appendix Erreur ! Source durenvoi introuvable.)

---

## Author Comment (AC1) · 11 Jun 2018

The authors would first like to express their gratitude to reviewer #1 for the careful comments on the work they have submitted for publication and the editor for the opportunity to improve the manuscript.

The actual aim of this article is not to describe and discuss the root cause of the noise of MERLIN system and the normality of the signal distributions, but rather to present and assess the biases that are produced by the averaging algorithms under the assumption of normal signal distribution (Qon,off). This assumption is not a rough approximation like it is often done, when one does not know better. It is justified by real measurements (out of the scope of the article) and also by theory, since for the

high number of photons (dark + signal approx. 1000) within the signal the Poisson statistics approximates (a shifted) Gaussian distribution already very well (central limit theorem). And the electronic part is also Gaussian because it is mainly thermal noise. The authors propose to add details in the article to precise the physical nature of the noise and justify the gaussian distribution approximation.

A confusion could arise from an unclear definition of what authors call “signals”. Qon and Qoff are called on-line and off-line signals in the article though they are not raw lidar signals (photo-electron count) measured by the detector. They are derived quantities from the backscattered signal (strictly positive count of photo-electrons by the detector). The received raw signals are the sum of the lidar signal and a background signal which is produced by background light, detector dark current and electronic offset. The computation of Qon and Qoff quantities from the raw signal includes an estimation of the energy of the backscatter signal and of the background signal. A subtraction of the background signal is then preformed which can lead to a negative value for Qon and Qoff (usually only Qon) for a very low target reflectivity. In the case where Qon is negative, it does not mean that no information is conveyed by the measurement but rather that the it is inaccessible due to the relatively high level of noise. Therefore, following the remarks of the reviewer #1, we propose to rename these quantities “calibrated signals” in the manuscript.

The fact that the manuscript highlights the mathematical aspects of bias correction algorithms is intended to insist on the generality of the approach. Their validity is then verified on the MERLIN system. The real noise parameters of MERLIN system used to simulate calibrated signals are taken into account via the simplified parametric equation giving the SNR from the reflectivity. The parameterization is deduced from instrumental characteristics and is provided by the sub-contractor (ASG) in charge of the development of the payload. The set of assumptions leading to this equation have not been detailed in the article though it could be interesting to explain it in appendix to the article. Figure 9 specifically shows this dependence of the on-line and off-line SNR

to the reflectivity. The SNR distributions used in the article are then indeed based on the real MERLIN characteristics. A sentence will be added to the article to make this fact clear. An appendix detailing the assumptions to derive the SNR from the reflectivity will be added. Furthermore, the detector noise being predominant, the successive measurements Qon and Qoff can be considered as independent.

Concerning the reference to Werle et al. (1993), there is indeed no justification of the statement that the bias caused by real system drift is negligible compared to averaging processing biases. Thus, this statement will be removed from the final version of manuscript. Anyway, real system drift is out of the scope of this article. This digression is introduced to clarify the position of the article so that no confusion is possible when the term “bias” is used.

Reviewer #1: “I may be missing the point here but I fail to understand how a higher threshold of usable signal before the computation of the DAOD could mitigate the fact that the Taylor bias correction does not succeed in quantifying the bias on any of the four mean reflectivity values. That is equivalent of excluding low SNR cases, which may be valid for data quality control, but does not mitigate the fact that the approach does not succeed in those cases.” In fact, when the threshold is set to be zero – which is the lowest mathematically possible value – we allow values down to this limit. A sample of Qon (hence SNRon estimate) that comes close to it would generate a large negative spike (1/SNRon$^2$) dominating all other values in the ensemble. Consequently, by choosing a higher threshold (strictly positive), we exclude the lower SNR cases and reach a better estimate of the bias for the remaining values. However, as noted by reviewer #1, when the SNR is low, the AVX and AVC methods does not succeed in estimating the XCH4 within the error specifications.

Furthermore, a table will be added to the appendix to show the results obtained by averaging quotients in order to quantify the bad performance of this scheme. A sentence will be added to clarify the distinction between DIAL and IPDA techniques. To avoid any confusion between bias and skewness, the expression “but skewed” will be

replaced by the expression “and present a bias” in the legend of Figure 3.

In addition to the modifications presented above, the authors will deal with minor comments as unclear syntax, grammar and spelling errors, numbering of figures, reference to relevant sources, broken links and numerical approximation consistency.

An updated version of the manuscript will soon be published following this response.

---

## Referee Comment (RC2) · Anonymous Referee #3 · 11 Aug 2018

The authors describe 4 different averaging schemes for implementing an Integrated Path Differential Absorption (IPDA) retrieval of methane, and provide mathematical analyses for 3 of them in regards to bias inherent in each approach. The context provided is the ESA MERLIN lidar mission, but the treatment provided is quite general and not specific to particular mission parameters.

I think the results provided are important considerations for developing the CH4 retrieval algorithms. Horizontal averaging is a necessity in the retrieval, especially for weak signals, and the authors address the ramifications of various algorithmic design choices in regard to the bias inherent in the different averaging approaches. Congratulations to the authors on sharing an interesting result. It would be interesting to revisit this treatment with real mission data after MERLIN launches or with other suborbital

data from mission formulation concepts like NASA's ASCENDS.

Isn't it true that in practice, a combination of the approaches presented might be necessary? For instance, the offline and online signals might need to be averaged separately first in order to accurately identify weak signals. Then further averaging of the DAOD or column mixing ratio can be applied to hammer down the noise.

I understand that the negative values indicated in Fig 3 are due to deriving the signal by subtracting a background value. For low SNR signals, the noise can dominate and push the background-corrected signal negative. In practice these cases would likely be filtered out by quality control executors, resulting in a skewed distribution as indicated. I would ask that the authors clarify what is meant by the negative signal values in the revised manuscript.

It is not obvious to me how or if the skewed distribution implies a bias. As I mentioned, in practice, I think such negative signal values would probably be filtered out so as not to enter the analysis. Please clarify.

Is laser speckle the dominant source of the statistical fluctuations? If so, speckle should be specifically treated in the manuscript.

Clouds in the field of view are a significant factor that are not treated. Partial/Spotty clouds might necessitate short averaging times to take measurements in the gaps. Thin cirrus clouds might be difficult to detect, yet cause significant biases. The authors should provide any input they might have on quantifying these factors.

Throughout the paper, the term "through" of the spectral line is used. Should this be "trough" or "center"? Furthermore, on page 2, line 8 is it really several absorption lines or just 1 selected methane line?

Page 2, line 14: "analyzes" should be "analysis"

---

## Author Comment (AC2) · 27 Aug 2018

The authors would like to express their gratitude to Anonymous Referee #3 (AR3) for the careful comments on the work they have submitted for publication and again the editor for the opportunity to improve the manuscript.

Indeed, the authors wanted the approach to be general and the article to study and suggest algorithms to minimize the bias when horizontally averaging IPDA lidar data. The MERLIN lidar mission, a CNES/DLR joint mission (not ESA), provides the variables needed for the simulation, however, the results are not only applicable to this specific mission and can be generalized to other ones. The authors agree with AR3 that it could be beneficial to revisit this treatment with real MERLIN data once the mission launched,

or for other missions. Nevertheless the study is performed here with simulated LIDAR signals derived from MERLIN mission parameters.

The use of a combined approach where the signals are averaged first before performing an averaging of DAOD or column mixing ratios does not seem to be the best method according to this study. The main reason why first averaging signals as much as possible is preferable is that the statistical bias correction for column averaging relies on the estimations of the on-line and off-line SNRs that are not perfect. When averaging signals, the SNR are averaged so that the variability is reduced and the statistical bias correction is better estimated. The simplified simulation performed on the three scenes shows that type 2 geophysical bias (due to the linearization of DAOD variations and the correlation of signal and transmission fluctuations) is very well estimated when the number of averaged shots is sufficient. A second reason not to perform a combined averaging scheme is that the averaging of signals produces an average XCH4 that is weighted by the off-line signal strength and thus take into account the relative information contained in each single signal. However, averaging DAOD or mixing ratios assigns uniform weights to each DAOD or column mixing ratio. Combining the two approaches would lead to a mixed weighting average XCH4 which can be confusing except if great care is taken when defining and using the respective weights.

As mentioned by AR3, "It is not obvious to me how or if the skewed distribution implies a bias." And indeed, a skewed distribution does not imply a bias (a PDF can be skewed and have zero bias). The reference to the skewness of the DAOD distribution when the signals are normally distributed is misleading in the manuscript. Such a DAOD distribution is generally biased and skewed but the important point that should have been highlighted is the bias of the distribution not its skewness. Therefore, all reference to the skewness of the DAOD distribution in the revised manuscript will be cautiously reviewed and highlights will be put on bias aspects.

As suggested, the manuscript will be revised to clarify what is meant by the negative signal values. First the term "signal" can bring confusion and will be replaced by

the term "calibrated signal" in the revised manuscript. The calibrated signal values can come to be negative as a treatment is performed to remove the background light power from the noisy measured signals. When we choose to filter out the negative calibrated signal values, as suggested by reviewer AR3, we deliberately chose to ignore the values that fall below the estimated background level even though they do convey information about the methane content of the column. Thus a positive bias appears due to the filtering process. By taking into account the filtering process (and general assumptions about the signal distribution) it is possible to correct this bias by introducing the truncated Gaussian distribution. The revised manuscript will develop some additional explanation about the bias produced by the filtering process.

In the typical conditions, the laser speckle is not the dominant source of the statistical fluctuation. The normality of the calibrated signal distributions is justified by real measurements (out of the scope of the article) and also by theory, since for the high number of photons (dark + signal approx. 1000) within the signal the Poisson statistics approximates (a shifted) Gaussian distribution already very well (central limit theorem). And the electronic part is also Gaussian because it is mainly thermal noise. The authors will add details in the article to precise the physical nature of the noise.

Clouds in the field of view have not been treated in the article as the goal was to explore and derive best algorithmic approaches to horizontally average with the least bias possible. The authors agree that the presence of opaque clouds can lead to a biased measurement as the contribution of hidden layers under the cloud would not be sounded by the instrument. Therefore, he processing chain of MERLIN mission includes a flagging protocol to separate between signals with or without opaque clouds. Two averaged XCH4 should be provided (clear-sky shots only and all shots). Concerning non-opaque clouds, as they would equally attenuate the off-line and on-line signals, they would only decrease the SNR but would not affect the measurement otherwise.

In addition to the modifications presented above, the authors will deal with minor comments (spelling errors).

---

## Author Response (AR1)

**Point-by-point response to all Referee Comments**

**Response to RC1 by AR1**

**Point 1: on measurement noise**

**Comment from AR1**

"In fact, the statistical bias induced by measurement noise is central to the analysis. However, that measurement noise is never defined and its characteristics are never described. Is it shot noise? Detector noise? Or some other "measurement" noise? Is it always purely a Gaussian, or a Poisson or some other distribution? What is the source of the "measurement noise"? These assumptions and questions are central to their analysis but are not well defined."

"The total noise contributions affecting off-line and on-line signals are statistically independent. Are they statistically independent? How do we know that? Can they be correlated if they have the same source? Does that change the results?"

"Furthermore, due to the relatively high number of photons in a single pulse, we can assume that these random variables are normally distributed around a mean value. There was no mention about the actual number of photons in a single pulse prior to this statement so we do not actually know how many photons are in the signals, Q. If we are photon (shot) noise limited the distribution should be Poisson not normal. Also, in the cases where the SNR is low (i.e. "the relatively high number of photons" is low) does the randomness assumption still hold?"

"Methodology to test averaging algorithms and their bias corrections How are the online and off-line signals computed purely from surface reflectivity and a random noise simulation? What assumptions were made for laser power, field of view, detector NEP etc etc.? Figure 8 fails to illuminate how Qon and Qoff are computed."

"Figure 9 shows the theoretical dependence of the SNR to the reflectivity according to instrument characteristics (assumptions) but those characteristics are never listed."

**Author's Response**

The aim of this article is not to describe and discuss the root cause of the noise of MERLIN system and the normality of the signal distributions, but rather to present and assess the biases that are produced by the averaging algorithms under the assumption that the signal noise follows a normal distribution (Qon, Qoff). Consequently, the authors decided not to be specific concerning the nature of the noise that affects the measurement. However, as it is important to justify the source of such noise as it causes averaging biases, the following modifications of the manuscript are proposed.

The fact that the manuscript highlights the mathematical aspects of bias correction algorithms is intended to insist on the generality of the approach. Their validity is then verified on the MERLIN system. The real noise parameters of the MERLIN system used to simulate calibrated signals are taken into account via the simplified parametric equation giving the SNR from the reflectivity. The parameterization is deduced from instrumental characteristics and is provided by the sub-contractor for the Instrument (Airbus Defence and Space GmbH) in charge of the development of the payload. The set of assumptions leading to this equation have not been detailed in the article though it could be interesting to explain it in an appendix to the article. Figure 9 specifically shows this dependence

of the on-line and off-line SNR to the reflectivity. The SNR distributions used in the article are then indeed based on the real MERLIN characteristics. A sentence will be added to the article to make this fact clear. An appendix detailing the assumptions to derive the SNR from the reflectivity will be added. Furthermore, the detector noise being predominant, the successive measurements Qon and Qoff can be considered as independent.

**Author's Change in Manuscript**
Creation of three subsections under section 2 one of which is called "MERLIN measurement noise" and describes the source of the noise of MERLIN measurement. Furthermore, an appendix called "Signal generation and noise model" has been added to describe the different factors that have been considered for the parametric noise model to generate the signals.

**Point 2: on DIAL and IPDA distinction**

**Comment form AR1**
"Introduction: This technique relies on the Differential Absorption Lidar (DIAL) measurements of a space-borne laser. This sentence is somewhat redundant considering the previous sentence: "…based on an Integrated Path Differential Absorption (IPDA) lidar. If the authors wish to make a distinction between IPDA and DIAL or further explain the technique then I suggest they should do so."

**Author's Response and Change in Manuscript**
A paragraph has been modified in the introduction of the revised manuscript to clarify the distinction between DIAL and IPDA lidar.

**Point 3: on real system drifts**

**Comment form AR1**
"The former bias being negligible compared to the latter. If I interpreted this statement correctly then the authors argue that: "…biases caused by real system drift…" are negligible compared to biases that are caused by the non-linearity of the IPDA lidar measurement equation. Is this justified by any experimental (or theoretical) evidence of the biases caused by real (MERLIN) system drift? I do not see how this statement is supported by any evidence. In fact, the reference cited earlier in the paragraph (Werle 1993) actually discussed biases caused by real system drifts."

**Author's Response**
Concerning the reference to Werle et al. (1993), there is indeed no justification of the statement that the bias caused by real system drift is negligible compared to averaging processing biases. Thus, this statement will be removed from the final version of manuscript. Anyway, real system drift is out of the scope of this article. This digression is introduced to clarify the position of the article so that no confusion is possible when the term "bias" is used.

**Author's Change in Manuscript**
The unjustified sentence have been removed from the revised manuscript.

**Point 4: on the negative valued signals**

**Comments from AR1**
"First, I do not understand how Qon can be negative."

"I fail to understand how either one or both lidar signals can take negative values. What does it physically mean for a lidar return signal to be negative? If the signal is completely absorbed or is very low (because of clouds or low reflectivity) then it can be zero but not negative."

"I do not agree with the statement: This can actually happen as Qon,off correspond to the photon count from which the background has been subtracted. I do not agree that this explains "negative" signals! It means there is no signal and all we have is background noise (could be solar background or detector noise). That does not make the lidar signal negative!"

"Figure 3: A significant part of on-line signals (orange) is negative – I do not understand how a signal can be negative. What part of Eq.(1) is negative for Qon? I could be missing something but it is not clear."

**Author's Response**

A confusion could arise from an unclear definition of what authors call "signals". Qon and Qoff are called on-line and off-line signals in the article though they are not raw lidar signals (photo-electron count) measured by the detector. They are derived quantities from the backscattered signal (strictly positive count of photo-electrons by the detector). The received raw signals are the sum of the lidar signal and a background signal which is produced by background light, detector dark current and electronic offset. The computation of Qon and Qoff quantities from the raw signal includes an estimation of the energy of the backscatter signal and of the background signal. A subtraction of the background signal is then preformed which can lead to a negative value for Qon and Qoff (usually only Qon) for a very low target reflectivity. In the case where Qon is negative, it does not mean that no information is conveyed by the measurement but rather that it is inaccessible due to the relatively high level of noise.

**Author's Change in Manuscript**

The ambiguous reference to "signal" has been replaced by "calibrated signals" in the manuscript when necessary. Plus, a paragraph has been added in "2.2 MERLIN processing chain" to explain how can calibrated signals be negative.

**Point 5: on Averaging of quotients**

**Comment from AR1**

"The fourth scheme AVQ is dismissed because it "gives very bad performance". It would have been much better to show the bad performance otherwise we just have to trust the authors."

**Author's Response and Change in Manuscript**

A table has been added to the appendix (Appendix B: Averaging of quotients) to show the results obtained by averaging quotients in order to quantify the bad performance of this scheme.

**Point 6: on the Threshold of usable signals**

**Comment from AR1**

"I may be missing the point here but I fail to understand how a higher threshold of usable signal before the computation of the DAOD could mitigate the fact that the Taylor bias correction does not succeed in quantifying the bias on any of the four mean reflectivity values. That is equivalent of excluding low SNR cases, which may be valid for data quality control, but does not mitigate the fact that the approach does not succeed in those cases."

**Author's Response**

In fact, when the threshold is set to be zero – which is the lowest mathematically possible value – we allow values down to this limit. A sample of Qon (hence SNRon estimate) that comes close to it would generate a large negative spike ($1/SNR_{on}^2$) dominating all other values in the ensemble. Consequently, by choosing a higher threshold (strictly positive), we exclude the lower SNR cases and reach a better estimate of the bias for the remaining values. However, as noted by reviewer #1,

when the SNR is low, the AVX and AVC methods does not succeed in estimating the XCH4 within the error specifications.

**Author's Change in Manuscript**
No change required.

**Point7: on minor mistakes**

**Comment from AR1**
"Abstract: This article discusses how to process horizontal averaging in order to avoid the bias caused by the non-linearity of the measurement equation with measurements affected by random noise and horizontal geophysical variability. This sentence is a bit confusing. Did the authors mean: This article discusses how to process this horizontal averaging in order to avoid the bias caused by the non-linearity of the measurement equation and measurements affected by random noise and horizontal geophysical variability?"

"To be precise: 0.07% of 1780 ppb is 1.25 ppb (see similar comment below)"

"Figure 2 is mentioned before Figure 1"

"I believe operational analyzes should be operational analyses (plural of the noun). Analyze (British spelling) is the verb."

"ASCENDS mission (NASA carbon dioxide IPDA lidar mission) – the appropriate reference listed in the references is needed. In addition, I believe it is best to define acronyms fully the first time they are used."

"1% of 1780ppb is 17.8 ppb and 0.2% is 3.6 ppb. I understand the authors round up or down but they just need to be consistent."

"Given the statement: the single shot on-line and off-line random error is reduced by a factor of sqrt(150)~12, I don't quite understand the example the authors gave: For instance, for the typical reflectivity (0.1), the on-line and off-line signal to noise ratios are of the order of 7 and 16 respectively (resp.) and the equivalent SNRs for the averaged signals are resp. 78 and 192. But sqrt(150)*7 =86 and sqrt(150)*16=196 not 78 and 192."

"This process greatly decrease the RRE should be: This process greatly decreases the RRE"

"Averaging schemes and bias correction: a theoretical approach The expected value of an random variable should be: The expected value of a random variable"

"…then derived from the pressure at every levels should be: …then derived from the pressure at every level."

"This sentence needs to be rewritten. It is confusing and not at all clear what the authors are trying to say: Consequently, this scheme is more sensitive to the less noisy measurements which, on the one hand, implies that the variance of average quantities is lower but, on the other hand, a correlation between methane concentration and reflectivity implies a bias."

"Link missing: (cf. Appendix Erreur ! Source du renvoi introuvable.)."

"This scheme gives very bad performances should be: This scheme gives very bad performance."

"Figure 1: On every averaging window, geophysical parameters altitude (or scattering surface elevation when there are clouds) or reflectivity vary. Should be: On every averaging window,

geophysical parameters such as altitude (or scattering surface elevation when there are clouds) and reflectivity vary"

"Link error: Statistical bias due to measurement noise mixed with geophysical biases into the non-linear equation (cf. Appendix Erreur ! Source du renvoi introuvable.)"

"Table 1: Link error: Statistical bias due to measurement noise mixed with geophysical biases into the non-linear equation (cf. Appendix Erreur ! Source du renvoi introuvable.)"

**Author's Response and Change in Manuscript**
In addition to the modifications presented above, the authors have dealt with minor comments as unclear syntax, grammar and spelling errors, numbering of figures, reference to relevant sources, broken links and numerical approximation consistency.

**Response to RC2 by AR3**
**Point 1: on the combined approach of averaging**
**Comment from AR3**
"Isn't it true that in practice, a combination of the approaches presented might be necessary? For instance, the offline and online signals might need to be averaged separately first in order to accurately identify weak signals. Then further averaging of the DAOD or column mixing ratio can be applied to hammer down the noise."

**Author's Response**
The use of a combined approach where the signals are averaged first before performing an averaging of DAOD or column mixing ratios does not seem to be the best method according to this study. The main reason why first averaging signals as much as possible is preferable is that the statistical bias correction for column averaging relies on the estimations of the on-line and off-line SNRs that are not perfect. When averaging signals, the SNR are averaged so that the variability is reduced and the statistical bias correction is better estimated. The simulation performed on the three scenes shows that type 2 geophysical bias (due to the linearization of DAOD variations and the correlation of signal and transmission fluctuations) is very well estimated when the number of averaged shots is sufficient. A second reason not to perform a combined averaging scheme is that the averaging of signals produces an average XCH4 that is weighted by the off-line signal strength and thus take into account the relative information contained in each single signal. However, averaging DAOD or mixing ratios assigns uniform weights to each DAOD or column mixing ratio. Combining the two approaches would lead to a mixed weighting average XCH4 which can be confusing for the users except if great care is taken when defining and using the respective weights.

**Author's Change in Manuscript**
No change required

**Point 2: on negative signal values and signal filtering**
**Comment from AR3**
"I understand that the negative values indicated in Fig 3 are due to deriving the signal by subtracting a background value. For low SNR signals, the noise can dominate and push the background-corrected signal negative. In practice these cases would likely be filtered out by quality control executors, resulting in a skewed distribution as indicated. I would ask that the authors clarify what is meant by the negative signal values in the revised manuscript."

**Author's Response and Change in Manuscript**

The calibrated signal values can come to be negative as a treatment is performed to remove the background light power from the noisy measured signals. When we choose to filter out the negative calibrated signal values, as suggested by reviewer AR3, we deliberately chose to ignore the values that fall below the estimated background level even though they do convey information about the methane content of the column. Thus a positive bias appears due to the filtering process. By taking into account the filtering process (and general assumptions about the signal distribution) it is possible to correct this bias by introducing the truncated Gaussian distribution (Eq. 16).

**Author's Change in Manuscript**

The manuscript has been revised to clarify what is meant by the negative signal values in the added paragraph in "2.2 MERLIN processing chain". Plus, the term "signal" can bring confusion and has been replaced by the term "calibrated signal" in the revised manuscript. The section "3.3.1.Statistical bias on AVX and AVD" has been modified to include some additional explanation about the bias produced by the filtering process.

**Point 3: on the skewness of the DAOD distribution**

**Comment from AR3**

"It is not obvious to me how or if the skewed distribution implies a bias. As I mentioned, in practice, I think such negative signal values would probably be filtered out so as not to enter the analysis. Please clarify."

**Author's Response**

The reference to the skewness of the DAOD distribution when the signals are normally distributed is misleading in the manuscript. Such a DAOD distribution is generally biased and skewed but the important point that should have been highlighted is the bias of the distribution not its skewness.

**Author's Change in Manuscript**

All references to the skewness of the DAOD distribution in the revised manuscript have been cautiously reviewed and highlights have been put on bias aspects.

**Point 4: on sources of statistical fluctuations**

**Comment from AR3**

"Is laser speckle the dominant source of the statistical fluctuations? If so, speckle should be specifically treated in the manuscript."

**Author's Response**

In the standard conditions of reflectivity (0.1 sr$^{-1}$), the dominant source of statistical fluctuations is the detector noise (mainly thermal). Other sources of bias can be neglected in the study.

**Author's Change in Manuscript**

Creation of three subsections under section 2 one of which is called "MERLIN measurement noise" and describe the source of the noise of MERLIN measurement. Furthermore, an appendix called "Signal generation and noise model" has been added to describe the different factors that have been considered for the parametric noise model to generate the signals.

**Point 5: on the procession of clouds**

**Comment from AR3**

"Clouds in the field of view are a significant factor that are not treated. Partial/Spotty clouds might necessitate short averaging times to take measurements in the gaps. Thin cirrus clouds might be

difficult to detect, yet cause significant biases. The authors should provide any input they might have on quantifying these factors."

**Author's Response**

The data processing of MERLIN will produce two separate products. The first one computes the average for clear-sky shots only, and the other one that averaged all shots.

**Author's Change in Manuscript**

A sentence has been added to the introduction of the revised manuscript to precise the data processing done in the presence of clouds.

**Point 6: on minor corrections**

**Comment from AR3**

"Throughout the paper, the term "through" of the spectral line is used. Should this be "trough" or "center"? Furthermore, on page 2, line 8 is it really several absorption lines or just 1 selected methane line?"

"Page 2, line 14: "analyzes" should be "analysis""

**Author's Response and Change in Manuscript**

In addition to the modifications presented above, the authors will deal with minor comments (spelling errors).

[revised manuscript text omitted]
^{\mathrm{off}} + \sigma^{\mathrm{off}} \cdot X^{\mathrm{off}}}{\mu^{\mathrm{on}} + \sigma^{\mathrm{on}} \cdot X^{\mathrm{on}}}\right) = \frac{1}{2} \cdot \ln\left(\frac{\mu^{\mathrm{off}}}{\mu^{\mathrm{on}}}\right) + \frac{1}{2} \cdot \ln\left(1 + \frac{X^{\mathrm{off}}}{SNR^{\mathrm{off}}}\right) - \frac{1}{2} \cdot \ln\left(1 + \frac{X^{\mathrm{on}}}{SNR^{\mathrm{on}}}\right), \tag{11}$$

where $X^{\mathrm{on,off}}$ follow standard normal distributions. And the signal-to-noise ratios are defined as:

15  $$SNR^{\mathrm{on,off}} = \frac{\mu^{\mathrm{on,off}}}{\sigma^{\mathrm{on,off}}}. \tag{12}$$

The first term of Eq. (11) is the parameter that needs to be estimated (i.e. the unbiased DAOD) and the two last terms are error terms that correspond to the bias of $\hat{\delta}$ due to the non-linearity of the function:

$$Bias_{\mathrm{stat}}(\hat{\delta}) = \frac{1}{2} \cdot E\left[\ln\left(1 + \frac{X^{\mathrm{off}}}{SNR^{\mathrm{off}}}\right)\right] - \frac{1}{2} \cdot E\left[\ln\left(1 + \frac{X^{\mathrm{on}}}{SNR^{\mathrm{on}}}\right)\right]. \tag{13}$$

The task is now to evaluate this bias to remove, or at least reduce it. Analytically, under the normal distribution hypothesis,

20  the expected values are defined by:

$$E\left[\ln\left(1 + \frac{X^{\mathrm{on,off}}}{SNR^{\mathrm{on,off}}}\right)\right] = \frac{1}{\sqrt{2\pi}} \int_{-SNR^{\mathrm{on,off}}}^{+\infty} \ln\left(1 + \frac{x}{SNR^{\mathrm{on,off}}}\right) \cdot e^{-\frac{x^2}{2}} \mathrm{d}x. \tag{14}$$

Providing that $SNR^{\mathrm{on,off}}$ are high enough, we can use a Taylor expansion of the logarithm around zero so that the bias can be approximated by the following formula (Bösenberg, 1998):

$$Bias_{\mathrm{stat}}(\hat{\delta}) \approx \frac{1}{4}\left[\frac{1}{(SNR^{\mathrm{on}})^2} - \frac{1}{(SNR^{\mathrm{off}})^2}\right]. \tag{15}$$

25  The assumption that the calibrated signals follow a normal distribution does not rigorously hold when the DAOD is computed. Indeed, over dark surfaces (low reflectivity), the SNR may happen to be so low that either one or both calibrated signals $Q^{\mathrm{off}}$ and $Q^{\mathrm{on}}$ takes negative values, hence, DAOD is undefined. This can actually happen as the calibrated signal $Q^{\mathrm{on,off}}$ is computed from the raw signal that correspond to athe photon count (positive quantity) from which the estimated background energy has been subtracted. Whenever one of the calibrated signals is negative, the corresponding couple $(Q^{\mathrm{off}}, Q^{\mathrm{on}})$ must be

discarded. And as the lowest calibrated signals are systematically discarded from the averaging set the average measurement is biased. This bias can be corrected by considering equations (11) and (13) with $X^{\mathrm{on,off}}$ as a left-truncated normal distribution with a mean value of zero, a variance of one and a left-truncation at $-SNR^{\mathrm{on,off}}$ (Johnson et al., 1994). When done so it comes that:

$$\quad E\left[\ln\left(1 + \frac{X^{\mathrm{on,off}}}{SNR^{\mathrm{on,off}}}\right)\right] = \frac{1}{\sqrt{2\pi}\left[1 - \Phi\left(-SNR^{\mathrm{on,off}}\right)\right]} \int_{-SNR^{\mathrm{on,off}}}^{+\infty} \ln\left(1 + \frac{x}{SNR^{\mathrm{on,off}}}\right) \cdot e^{-\frac{x^2}{2}} \mathrm{d}x \,, \tag{16}$$

where $\Phi$ is the standard normal cumulative distribution function.

To correct the bias due to the non-linearity of the IPDA lidar equation, the SNR must be estimated. Once done, the bias correction scheme would either need to estimate the bias directly from the approximate Taylor expansion formula of Eq. (15) or estimate the bias using Eq. (13) and a numerical computation of Eq. (16). Typically, for MERLIN observations, the error made by the Taylor expansion of Eq. (15) instead of using Eq. (16) is lower than 1 ppb on the $X_{CH_4}$ for a surface reflectivity value greater than 0.1 ($SNR^{\mathrm{off}} \approx 16$ and $SNR^{\mathrm{on}} \approx 7$) as shown on Figure 4. Table 2 shows the error made by using the Taylor expansion instead of computing the truncated normal integral. For values of reflectivity smaller than 0.1, it would be preferable to use the exact formula for the bias presented in equations (13) and (16). Further study (not presented here) shows that for very low reflectivity, the estimation of the noise induced bias is really sensitive to an error on the SNR and this correction is no longer applicable in practice. The way statistical bias on the DAOD is translated to bias on $X_{CH_4}$ will be treated in section 3.5.

**3.3.2 Statistical bias on AVS**

The third averaging scheme defined on Table 1, AVS, averages on-line and off-line calibrated signals separately. The corresponding estimator of the average DAOD is written:

$$\quad \hat{\delta}^{avs} = \frac{1}{2} \cdot \ln\left(\frac{\langle Q^{\mathrm{off}}\rangle}{\langle Q^{\mathrm{on}}\rangle}\right). \tag{17}$$

Consistently with section 3.3.1, we consider the individual calibrated signals to be normal random variables of mean $\mu_i^{\mathrm{on,off}}$ and standard deviation $\sigma_i^{\mathrm{on,off}}$. The parameters of the distributions depend on the shot $i$ since each shot is considered as the realization of a different distribution depending on the geophysical parameters of the scene (reflectivity, atmospheric transmission, surface pressure). The successive measurements are considered independent and, as the sum of independent normal random variables, is a normal random variable. We introduce $S^{\mathrm{on,off}}$ the average random variable:

$$S^{\mathrm{on,off}} = \langle Q^{\mathrm{on,off}}\rangle \sim \mathcal{N}\left(m^{\mathrm{on,off}}, \left(\epsilon^{\mathrm{on,off}}\right)^2\right), \tag{18}$$

where the mean and variance of $S^{\mathrm{on,off}}$ are:

$$m^{\mathrm{on,off}} = \langle \mu^{\mathrm{on,off}}\rangle, \tag{19}$$

[revised manuscript text omitted]

– SNR estimation
– Bias evaluation:
Option 1: Eq. (33) and (15)
Option 2: Eq. (33), (16) and (13) | – Discard negative signals
– SNR estimation
– Bias evaluation:
Option 1: Eq. (32) and (15)
Option 2: Eq. (32), (16) and (13) | – SNR estimation
– Equivalent window SNR by Eq. (21)
– Bias evaluation:
Option 1: Eq. (34) and (15)
Option 2: Eq. (34), (16) and (13) |
| **Geophysical bias evaluation** | – None (Type 1 geophysical bias built-in $w[IWF]$ weights of Table 1, line 1 and Eq. (33)) | – None | – Type 2 geophysical bias of Eq. (35)
– Type 3 geophysical bias of Eq. (36) not estimated |

Table 5: Computational details about averaging schemes and bias evaluation

[revised manuscript text omitted]